# OLMoE: Open Mixture-of-Experts Language Models

**Niklas Muennighoff**[ca]  **Luca Soldaini**[a]  **Dirk Groeneveld**[a]  **Kyle Lo**[a]  **Jacob Morrison**[a]
**Sewon Min**[a]  **Weijia Shi**[w]  **Pete Walsh**[a]  **Oyvind Tafjord**[a]  **Nathan Lambert**[a]
**Yuling Gu**[a]  **Shane Arora**[a]  **Akshita Bhagia**[a]  **Dustin Schwenk**[a]  **David Wadden**[a]
**Alexander Wettig**[ap]  **Binyuan Hui**  **Tim Dettmers**[a]  **Douwe Kiela**[c]  **Ali Farhadi**[aw]
**Noah A. Smith**[aw]  **Pang Wei Koh**[aw]  **Amanpreet Singh**[c]  **Hannaneh Hajishirzi**[aw]
[a] Allen Institute for AI  [c] Contextual AI  [w] University of Washington  [p] Princeton University
n.muennighoff@gmail.com  hannah@allenai.org

## Abstract

We introduce **OLMoE**,[1] a fully open, state-of-the-art language model leveraging sparse Mixture-of-Experts (MoE). **OLMoE-1B-7B** has 7 billion (B) parameters but uses only 1B per input token. We pretrain it on 5 trillion tokens and further adapt it to create **OLMoE-1B-7B-Instruct**. Our models outperform all available models with similar active parameters, even surpassing larger ones like Llama2-13B-Chat and DeepSeekMoE-16B. We present novel findings on MoE training, define and analyze new routing properties showing high specialization in our model, and open-source all our work: model weights, training data, code, and logs.

|  |  |  |
|---|---|---|
| 🤗 | **Weights** | https://hf.co/allenai/OLMoE-1B-7B-0924 |
|  | **Data** | https://hf.co/datasets/allenai/OLMoE-mix-0924 |
| ⭘ | **Code** | https://github.com/allenai/OLMoE |
| ⋮⋮ | **Logs** | https://wandb.ai/ai2-llm/olmoe/reports/ OLMoE-1B-7B-0924--Vmlldzo4OTcyMjU3 |

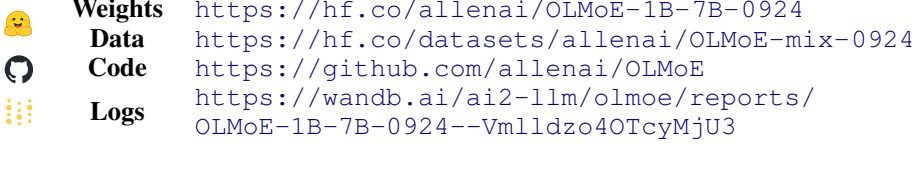

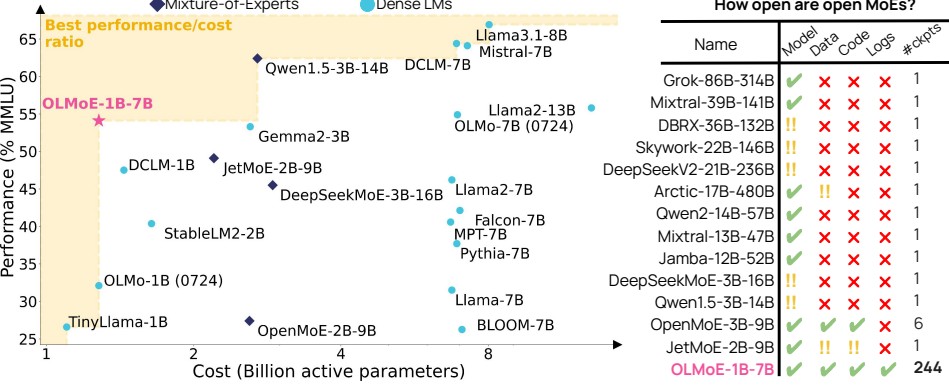

Figure 1: **Performance, cost, and degree of openness of open MoE and dense LMs.** Model names contain rounded parameter counts: `model-active-total` for MoEs and `model-total` for dense LMs. `#ckpts` is the number of intermediate checkpoints available. We highlight MMLU as a summary of overall performance; see §3 for more results. **OLMoE-1B-7B** performs best among models with similar active parameter counts and is the most open MoE.

## 1 Introduction

Despite significant advances in Large Language Models (LMs) on various tasks, there remains a clear trade-off between performance and cost in both training and inference. High-performing LMs are inaccessible for many academics and open-source developers as they are prohibitively expensive to

---

[1] This paper describes the first **OLMoE** from 09/2024. See Appendix L for an overview of a newer version.

build and deploy.[2] One approach to improve the cost-performance trade-off lies in using sparsely-activated Mixture-of-Experts (MoEs) (Shazeer et al., 2017). MoEs have several experts in each layer, only a subset of which is activated at a time (see Figure C1). This makes MoEs significantly more efficient than dense models with a similar number of total parameters, which activate all parameters for every input (Yun et al., 2024). For this reason, industry frontier models use MoEs including Gemini-1.5 (Team et al., 2024a) and reportedly GPT-4 (Chintala, 2024).

Most MoE models, however, are closed-source: While some have publicly released model weights (DeepSeek-AI et al., 2024b; Jiang et al., 2024; Shen et al., 2024; Team et al., 2024d; Team, 2024b), they offer limited to no information about their training data, code, or recipes (see Figure 1). While there have been prior efforts to make language modeling research fully accessible (Biderman et al., 2023; Groeneveld et al., 2024; Li et al., 2024a; Liu et al., 2023; Workshop et al., 2023; Zhang et al., 2024a) discussed in detail in Appendix A, they have been largely limited to dense LMs. This comes despite MoEs requiring *more* openness as they add complex new design questions to LMs, such as how many total versus active parameters to use, whether to use many small or few large experts, if experts should be shared, and what routing algorithm to use. The lack of open resources and findings about these details prevents the field from building cost-efficient open MoEs that approach the capabilities of closed-source frontier models.

To address these issues, we introduce **OLMoE**, a fully open Mixture-of-Experts language model with state-of-the-art performance among similarly-sized models. In particular, we pretrain **OLMoE-1B-7B** for 5.1 trillion tokens with 6.9B total parameters, of which only 1.3B are activated for each input token. This leads to a similar inference cost as using dense models with around 1B parameters, such as OLMo 1B (Groeneveld et al., 2024) or TinyLlama 1B (Zhang et al., 2024b), but requires more GPU memory to store its 7B total parameters. Our experiments show that MoEs train ~2× faster than dense LMs with equivalent active parameters (Figure 2). In Figure 1, we show that **OLMoE-1B-7B** significantly outperforms all open 1B models and displays competitive performance to dense models with significantly higher inference costs and memory storage (e.g., similar MMLU scores to Llama2-13B, which is ~10× more costly). Via instruction- and preference tuning, we create **OLMoE-1B-7B-INSTRUCT**, which we find exceeds various larger instruct models including Llama2-13B-Chat (Touvron et al., 2023b), OLMo-7B-Instruct (0724), and DeepSeekMoE-16B (DeepSeek-AI et al., 2024a) on common benchmarks (MMLU, GSM8k, HumanEval, etc.).

Our comprehensive set of controlled experiments highlights key design choices for MoEs (see Table 1) and LMs in general. One critical design decision for making MoEs performant is using fine-grained routing with granular experts (DeepSeek-AI et al., 2024a): we employ 64 small experts in each layer with 8 being activated. The choice of routing algorithm is also important: we find dropless (Gale et al., 2022) token-based routing (Shazeer et al., 2017) outperforms expert-based routing (Zhou et al., 2022). Our findings also include those that challenge prior work, such as the ineffectiveness of shared experts (DeepSeek-AI et al., 2024a) and the limited benefits of sparsely upcycling a pretrained dense LM into an MoE (Komatsuzaki et al., 2023) unless under small compute budgets. Finally, we present novel ways to analyze routing behavior in Mixture-of-Experts finding that for **OLMoE-1B-7B** routing saturates early in pretraining, experts are rarely co-activated, and experts exhibit domain and vocabulary specialization. We intend our fully open MoE to facilitate more research and analysis to improve our understanding of these models. We release training code, intermediate checkpoints (every 5000 steps), training logs, and training data under open-source licenses (Apache 2.0 http://www.apache.org/licenses/LICENSE-2.0 or ODC-By 1.0 https://opendatacommons.org/licenses/by/1-0/).

## 2 PRETRAINING AND ADAPTATION

**Pretraining** **OLMoE** is a decoder-only LM consisting of $N_L$ transformer (Vaswani et al., 2023) layers. The feedforward network (FFN) in dense models like OLMo (Groeneveld et al., 2024) is replaced with an MoE module consisting of $N_E$ smaller FFN modules called experts, of which a subset of $k$ experts is activated for each processed input token $x$ (also see Figure C1):

$$\text{MoE module}(x) = \sum_{i \in \text{Top-}k(r(x))} \text{softmax}\left(r(x)\right)_i E_i(x) \tag{1}$$

---

[2]For example, even with 16 H100 GPUs and several optimizations, Llama 3 405B only achieves a decoding throughput of around 100 tokens per second (Dubey et al., 2024).

Table 1: **Key MoE design choices and our setup for OLMoE-1B-7B based on our experiments.** Full configuration for **OLMoE-1B-7B** is in Appendix C.

| Design choice | Description | Experiment | OLMoE-1B-7B |
|---|---|---|---|
| Active params | # active parameters per input token | §4.1 | 1.3B active |
| Total params | Total # of parameters in the model | §4.1 | 6.9B total |
| Expert granularity | Using fine-grained small experts vs. a few large experts (Dai et al., 2024) | §4.2 | 64 small experts with 8 activated |
| Routing algorithm | How inputs are assigned to experts, e.g., a per token basis (e.g., 2 experts per token) or per expert basis (e.g., 2 tokens per expert), and if all tokens get assigned or some get dropped | §4.3 | Dropless (Gale et al., 2022) MoE with token choice |
| Expert sharing | Whether to share experts (Dai et al., 2024) | §B.1.1 | No shared expert |
| Sparse upcycling | Whether to start from a dense model (Komatsuzaki et al., 2023; Zhang et al., 2024c) | §B.1.2 | Not used |
| Load balancing loss | Auxiliary loss to penalize unequal assignment to experts harming performance (Shazeer et al., 2017) | §B.1.3 | Used with weight 0.01 |
| Router z-loss | Auxiliary loss to penalize large router logits that may cause instabilities (Zoph et al., 2022) | §B.1.4 | Used with weight 0.001 |

where $r$, called the router, is a learned linear layer mapping from the input logits to the chosen $k$ experts. A softmax is applied to the router outputs to compute routing probabilities for all $N_E$ experts. Each selected expert $E_i$ processes the input $x$, the output of which is then multiplied with its respective routing probability. The results are then summed across all chosen Top-$k$ experts to constitute the output of the MoE module for a single layer of the model out of its $N_L$ total layers. Key decisions in designing an MoE model include determining the number of activated and total parameters, the design of the experts (e.g., granularity, whether or not to include shared experts), and the choice of the routing algorithm. Moreover, training an MoE model can involve initializing from a dense model (sparse upcycling) and changing the training objective, such as including auxiliary load balancing and router z-losses. We run experiments to investigate each of these design choices in isolation in §4 and §B.1. We summarize our final decisions in Table 1: We use 1.3B active parameters out of a total of 6.9B, with 8 activated experts out of 64 per layer. We use dropless token choice routing (Gale et al., 2022): For each input token, the learned router network determines 8 experts to process it. We train **OLMoE-1B-7B** from scratch with two auxiliary losses: load balancing loss ($\mathcal{L}_{LB}$) (Shazeer et al., 2017) and router z-loss ($\mathcal{L}_{RZ}$) (Zoph et al., 2022), which we define and experiment with in §B.1.3 and §B.1.4, respectively. We multiply them with respective loss weights, $\alpha$ and $\beta$, and sum them linearly with the cross entropy loss ($\mathcal{L}_{CE}$) to arrive at our final training loss:

$$\mathcal{L} = \mathcal{L}_{CE} + \alpha\mathcal{L}_{LB} + \beta\mathcal{L}_{RZ} \tag{2}$$

For our pretraining data, we mix data from DCLM (Li et al., 2024a) and Dolma 1.7 (Soldaini et al., 2024), which includes: (1) a quality-filtered subset of Common Crawl, referred to as DCLM-Baseline, (2) StarCoder, Algebraic Stack and arXiv, used in both DCLM and Dolma 1.7, and (3) peS2o and Wikipedia from Dolma 1.7. We refer to our pretraining dataset as **OLMoE-MIX**. We train for a total of 5.133T tokens (1.3 epochs following Muennighoff et al. (2023b)) and provide data statistics in Table C1. Our full pretraining configuration for **OLMoE-1B-7B** is in Appendix C.

**Adaptation** We create **OLMoE-1B-7B-INSTRUCT** by following a standard adaptation recipe split into **instruction tuning** (Mishra et al., 2022; Wei et al., 2022; Sanh et al., 2022; Shen et al., 2023a; Zadouri et al., 2023) followed by **preference tuning** (Christiano et al., 2023; Bai et al., 2022; Rafailov et al., 2023) building on prior open models (Tunstall et al., 2023; Ivison et al., 2023; Wang et al., 2023). In our instruction tuning dataset, we add more code and math data to boost performance on downstream coding and math applications. Other models, such as GPT-4 (OpenAI et al., 2023) and Llama 3 (Dubey et al., 2024) similarly include samples from math datasets like GSM8k (Cobbe et al., 2021) or MATH (Hendrycks et al., 2021b) during pretraining. We also include No Robots and a subset of Daring Anteater as they are of high quality and add diversity, two key factors for successful

adaptation (Wang et al., 2023; Zhou et al., 2023a; Longpre et al., 2023a; Muennighoff et al., 2023a). We describe our adaptation datasets in Table C2 and hyperparameters in Appendix C.

## 3 RESULTS

Our evaluation procedure consists of three parts: **During pretraining** (Appendix F), **After pretraining**, and **After adaptation**. We detail the setup for each in Appendix D.

Table 2: **OLMOE-1B-7B after pretraining.** We compare with LMs of similar active parameters (1B, approximating speed and cost) or total parameters (7B, approximating memory). Model names include rounded parameter counts: `model-active-total` for MoEs and `model-total` for dense LMs (leading to differences from official names, e.g., "Gemma2-2B" has 2.6B active and total parameters (Team et al., 2024c)). We run all evaluations ourselves with 5 few-shots (Appendix D).

| | Active params | Open Data | MMLU | Hella-Swag | ARC-Chall. | ARC-Easy | PIQA | Wino-Grande |
|---|---|---|---|---|---|---|---|---|
| LMs with ~7-9B active parameters | | | | | | | | |
| Llama2-7B | 6.7B | ✗ | 46.2 | 78.9 | 54.2 | 84.0 | 77.5 | 71.7 |
| OLMo-7B (0724) | 6.9B | ✔ | 54.9 | 80.5 | 68.0 | 85.7 | 79.3 | 73.2 |
| Mistral-7B | 7.3B | ✗ | 64.0 | 83.0 | 78.6 | 90.8 | 82.8 | 77.9 |
| DCLM-7B | 6.9B | ✔ | 64.4 | 82.3 | 79.8 | 92.3 | 80.1 | 77.3 |
| Llama3.1-8B | 8.0B | ✗ | 66.9 | 81.6 | 79.5 | 91.7 | 81.1 | 76.6 |
| Gemma2-9B | 9.2B | ✗ | **70.6** | **87.3** | **89.5** | **95.5** | **86.1** | **78.8** |
| LMs with ~2-3B active parameters | | | | | | | | |
| OpenMoE-3B-9B | 2.6B | ✔ | 27.4 | 44.4 | 29.3 | 50.6 | 63.3 | 51.9 |
| StableLM-2B | 1.6B | ✗ | 40.4 | 70.3 | 50.6 | 75.3 | 75.6 | 65.8 |
| DeepSeek-3B-16B | 2.9B | ✗ | 45.5 | 80.4 | 53.4 | 82.7 | 80.1 | **73.2** |
| JetMoE-2B-9B | 2.2B | ✗ | 49.1 | **81.7** | 61.4 | 81.9 | 80.3 | 70.7 |
| Gemma2-3B | 2.6B | ✗ | 53.3 | 74.6 | 67.5 | 84.3 | 78.5 | 71.8 |
| Qwen1.5-3B-14B | 2.7B | ✗ | **62.4** | 80.0 | **77.4** | **91.6** | **81.0** | 72.3 |
| LMs with ~1B active parameters | | | | | | | | |
| Pythia-1B | 1.1B | ✔ | 31.1 | 48.0 | 31.4 | 63.4 | 68.9 | 52.7 |
| OLMo-1B (0724) | 1.3B | ✔ | 32.1 | 67.5 | 36.4 | 53.5 | 74.0 | 62.9 |
| TinyLlama-1B | 1.1B | ✔ | 33.6 | 60.8 | 38.1 | 69.5 | 71.7 | 60.1 |
| Llama3.2-1B | 1.2B | ✗ | 38.2 | 67.3 | 43.5 | 71.6 | 73.7 | 62.5 |
| DCLM-1B | 1.4B | ✔ | 48.5 | 75.1 | 57.6 | 79.5 | 76.6 | 68.1 |
| **OLMOE-1B-7B** | 1.3B | ✔ | **54.1** | **80.0** | **62.1** | **84.2** | **79.8** | **70.2** |

**After pretraining** In Table 2 we benchmark **OLMOE-1B-7B** on common downstream tasks. We find that **OLMOE-1B-7B** performs best among models that use less than 2B active parameters, making it the most economical option for many use cases of LMs. For larger budgets, Qwen1.5-3B-14B has stronger performance but has more than double the active and total parameters than **OLMOE-1B-7B**. We find that despite requiring ~6–7× less compute per forward pass, **OLMOE-1B-7B** outperforms some dense LMs with 7B parameters such as Llama2-7B (Touvron et al., 2023b), but falls short of others like Llama3.1-8B (Dubey et al., 2024). Figure 1 compares MMLU performance with active parameters, a proxy for the value of a model given its cost, of **OLMOE-1B-7B** and other LMs. **OLMOE-1B-7B** is the state of the art in its cost regime.

**After adaptation** In Table 3, we benchmark our instruction (SFT) and preference (DPO) tuning of **OLMOE-1B-7B**. SFT improves our model on all tasks measured. We observe a >10× gain on GSM8k, likely due to our inclusion of additional math data to account for the relatively small amounts of math data during pretraining (§2). DPO helps on most tasks, especially AlpacaEval. Our DPO model, which we refer to as **OLMOE-1B-7B-INSTRUCT**, has the highest average among all models benchmarked. We find it to outperform the chat version of Qwen1.5-3B-14B despite Qwen

Table 3: **OLMOE-1B-7B after adaptation.** Model names contain rounded parameter counts: `model-active-total` for MoEs and `model-total` for dense LMs. We run all evaluations ourselves (Appendix D). Models use different data mixes and setups for adaptation.

| Task (→)
Setup (→)
Metric (→) | MMLU
0-shot
EM | GSM8k
8-shot CoT
EM | BBH
3-shot
EM | Human-
Eval
0-shot
Pass@10 | Alpaca-
Eval 1.0
0-shot
%win | XSTest
0-shot
F1 | IFEval
0-shot
Loose Acc | Avg |
|---|---|---|---|---|---|---|---|---|
| OLMo-1B (0724) | 25.0 | 7.0 | 22.5 | 16.0 | - | 67.6 | 20.5 | - |
| +SFT | 36.0 | 12.5 | 27.2 | 21.2 | 41.5 | 81.9 | 26.1 | 35.9 |
| +DPO | 36.7 | 12.5 | 30.6 | 22.0 | 50.9 | 79.8 | 24.2 | 37.4 |
| OLMo-7B (0724) | 50.8 | 32.5 | 36.9 | 32.3 | - | 80.8 | 19.6 | - |
| +SFT | 54.2 | 25.0 | 35.7 | 38.5 | 70.9 | 86.1 | 39.7 | 49.3 |
| +DPO | 52.8 | 9.0 | 16.6 | 35.0 | 83.5 | **87.5** | 37.9 | 49.1 |
| JetMoE-2B-9B | 45.6 | 43.0 | 37.2 | 54.6 | - | 68.2 | 20.0 | - |
| +SFT | 46.1 | 53.5 | 35.6 | 64.8 | 69.3 | 55.6 | 30.5 | 50.4 |
| DeepSeek-3B-16B | 37.7 | 18.5 | 39.4 | 48.3 | - | 65.9 | 13.5 | - |
| +Chat | 48.5 | 46.5 | **40.8** | **70.1** | 74.8 | 85.6 | 32.3 | 57.0 |
| Qwen1.5-3B-14B | **60.4** | 13.5 | 27.2 | 60.2 | - | 73.4 | 20.9 | - |
| +Chat | 58.9 | **55.5** | 21.3 | 59.7 | 83.9 | 85.6 | 36.2 | 57.3 |
| **OLMOE-1B-7B** | 49.8 | 3.0 | 33.6 | 22.4 | - | 59.7 | 16.6 | - |
| **+SFT** | 51.4 | 40.5 | 38.0 | 51.6 | 69.2 | 84.1 | 43.3 | 54.0 |
| **+DPO** | 51.9 | 45.5 | 37.0 | 54.8 | **84.0** | 82.6 | **48.1** | **57.7** |

having >2× more parameters and its pretrained model outperforming **OLMOE-1B-7B** in Table 2. The 84% score on AlpacaEval also outperforms much larger dense models on the leaderboard (Li et al., 2023b), such as Llama2-13B-Chat (Touvron et al., 2023b).

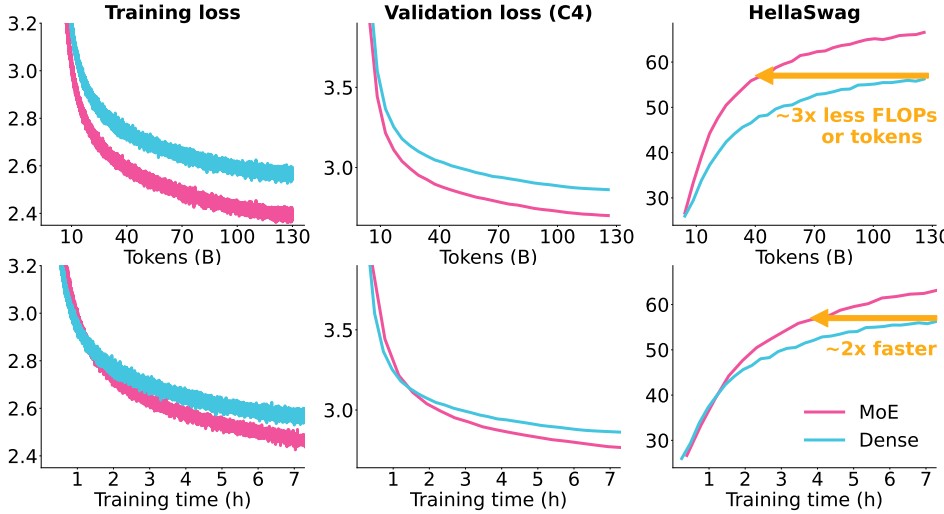

Figure 2: **MoE vs. Dense.** We train a 1.3B parameter dense model and a 1.3B active, 6.9B total parameter MoE model, each on 128 H100 GPUs. Apart from MoE-related changes, we train both with the same configuration for 130B tokens. The MoE has 64 experts per layer, 8 of which are activated with an FFN dimension of 1,024. The dense model has an FFN dimension of 8,192. Thus both have the same number of active parameters. **Top:** The MoE reaches the final dense performance with ~3× fewer tokens (or FLOPs, as both have the same active parameters ignoring the trivial router parameters). **Bottom:** Due to some memory overhead, this equates to ~2× faster training. More results, logs, and configurations: `https://wandb.ai/ai2-llm/olmoe/reports/Plot-MoE-vs-Dense--Vmlldzo4OTM0Mjkx`

## 4 EXPERIMENTING WITH ALTERNATIVE DESIGN CHOICES

This section contains some experiments that led to **OLMoE-1B-7B** with many more in Appendix B.

### 4.1 MIXTURE-OF-EXPERTS VS. DENSE

Prior work reports various speed-ups of MoEs over dense models: Artetxe et al. (2022) state MoEs require 2–4× less compute to match dense models, MoMa (Lin et al., 2024b) exhibits 2.6× FLOP savings for language tasks, Arctic (Snowflake, 2024b) yields 4× FLOP savings but for very different dense and MoE configurations, and Switch Transformer MoEs (Fedus et al., 2022) train 2-7× faster but for encoder-decoder models while the other works study decoder-only LMs (Radford et al., 2019).

In Figure 2, we compare MoEs and dense models in a controlled setup. We find that our MoE reaches the performance of the dense model with ~3× fewer tokens equivalent to ~3× less compute measured in FLOPs. However, due to the additional memory overhead of training the MoE with its 7B total parameters, it processes fewer tokens per second than the dense model (23,600 tokens per second per GPU for the MoE vs. 37,500 for dense). Thus, in terms of training time, it reaches the performance of the dense model only ~2× faster. There are likely optimizations possible that would bring the speed-up closer to the 3× token speed-up, which we leave to future work. Based on these results, we select an MoE configuration with 6.9B total and 1.3B active parameters matching OLMo-7B in total and OLMo-1B in active parameter count, respectively (Groeneveld et al., 2024). We provide more reasons for this configuration in Appendix J.

### 4.2 EXPERT GRANULARITY

Dai et al. (2024) propose to use small fine-grained experts to allow more combinations of experts and thus make the model more flexible. For example, the Mixtral model (Jiang et al., 2024) uses the common configuration of 8 experts per layer, 2 of which are activated. This allows for $\binom{8}{2} = 28$ combinations per layer. By halving the size of each expert and therefore doubling the number of experts to maintain the same compute and parameter budget, we can increase the possible combinations to $\binom{16}{4} = 1,820$. Krajewski et al. (2024) investigate compute-optimal granularity configurations finding that higher compute budgets warrant more granular experts.

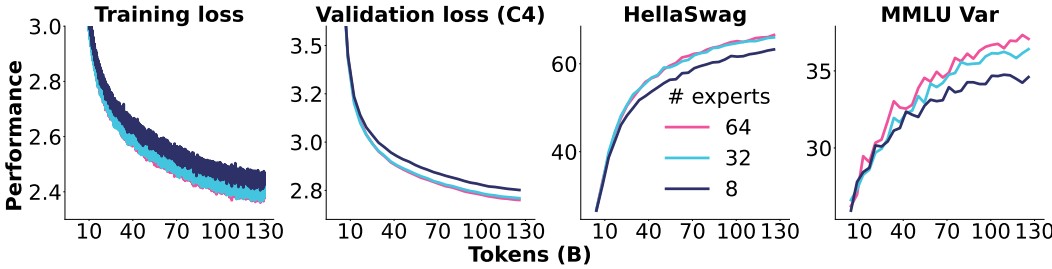

Figure 3: **Expert granularity.** We vary the number of experts in tandem with the FFN dimension to ensure that active and total parameters and thus compute cost remain the same. For example, for 64 experts, the FFN dimension is 1,024 and 8 experts are active, while for 32 experts it is 2,048 with 4 active experts. More results, logs, and configurations: https://wandb.ai/ai2-llm/olmoe/reports/Plot-Granularity--Vmlldzo4OTIxOTE4

In Figure 3, we observe that more granular experts improve training loss, validation loss, and downstream performance. The 8-expert configuration uses 1 active expert, which yields $\binom{8}{1} = 8$ combinations. By quartering the size of each expert but increasing the number to 32 with 4 active ones ($\binom{32}{4} = 35,960$ combinations), we observe an improvement of around 10% on HellaSwag and MMLU at around 130 billion tokens. However, we find that there are diminishing returns to granularity. The additional increase to 64 experts with 8 active ones ($\binom{64}{8} = 4,426,165,368$ combinations) improves downstream metrics by a smaller amount of 1–2%. For our **OLMoE-1B-**

**7B** compute budget[3] of $3 \times 10^{22}$, Krajewski et al. (2024) predict an optimal number of experts of 256 ($G = 32$ in their paper). However, their predictions are for compute-optimal models (Hoffmann et al., 2022; Clark et al., 2022), while we train for 5T tokens, which is orders of magnitude beyond what would be conventionally considered optimal for our model size. Thus, their predictions may not extend to our setup, and we stick with 64 experts for **OLMoE-1B-7B**, also due to the diminishing returns in Figure 3.

### 4.3 EXPERT CHOICE VS. TOKEN CHOICE

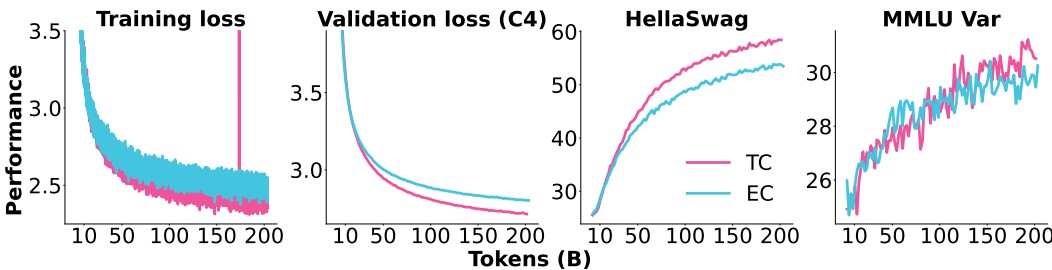

Figure 4: **Expert choice (EC) vs. token choice (TC).** Both models have an 8-expert MoE in every 2nd layer. For TC, 2 experts are activated per token, while for EC the capacity factor is 2. Thus, both models use the same number of active parameters. More results, logs, and configurations: `https://wandb.ai/ai2-llm/olmoe/reports/Plot-EC-vs-TC--Vmlldzo4MzkzMDM3`

The MoE router determines which experts process each input token (§2). There are two common types (Liu et al., 2024b): **expert choice (EC)** (Zhou et al., 2022) and **token choice (TC)** (Shazeer et al., 2017). For EC, each expert selects a fixed number of tokens per sequence. By design, this leads to each expert processing the same number of tokens. This is the main benefit of EC as it ensures perfect load balance, which improves training throughput and removes the need for a load balancing loss. The main issue of EC is that it is impractical for autoregressive generation where a single token is processed at each step rather than the entire sequence in one (Raposo et al., 2024). Another potential downside is EC can lead to token dropping, where some tokens are not selected by any expert, which can hurt performance (Gale et al., 2022). At the same time, it can lead to some tokens being processed by multiple experts, which could also be beneficial as it allows the model to allocate more compute to some tokens (Zhou et al., 2022). For TC, each token selects a fixed number of experts. This can lead to many tokens choosing the same expert, hurting training efficiency. Thus, TC is often used with a load balancing loss (Shazeer et al., 2017) to encourage equal distribution.

In Figure 4, we benchmark EC and TC. We find that TC outperforms EC for the same token budget for all tasks depicted. While Zhou et al. (2022) find EC to be better, our configuration slightly differs in that we use dropless MoEs (Gale et al., 2022) with a load balancing loss. Thus, our TC variant is expected to perform better than the TC variant in Zhou et al. (2022). We confirm findings that EC runs around 20% faster at 29,400 tokens per second per device versus 24,400 for TC (Zhou et al., 2022). EC may be more beneficial in a multimodal setup (Lin et al., 2024b) as dropping noisy image tokens is likely less harmful than text tokens. Thus, while we stick with TC for this release of **OLMoE**, we may revisit EC for future multimodal models.

## 5 MoE ANALYSIS

By advancing open and cost-efficient models (§1), **OLMoE-1B-7B** enables new research into LMs and MoEs. Making use of our released intermediate checkpoints, data, and code, we define and analyze four properties specific to MoEs: Router saturation (§5.1), Expert co-activation (§5.2), Domain specialization (§5.3), and Vocabulary specialization (§5.4).

### 5.1 ROUTER SATURATION

Router saturation, as a function of time $t$, represents the proportion of overlapping activated experts between the final checkpoint and an intermediary checkpoint at time $t$. Router saturation thus

---

[3]Approximated via $6 * N * D$ (Kaplan et al., 2020) with active parameters $N$ (1B) and tokens $D$ (5T).

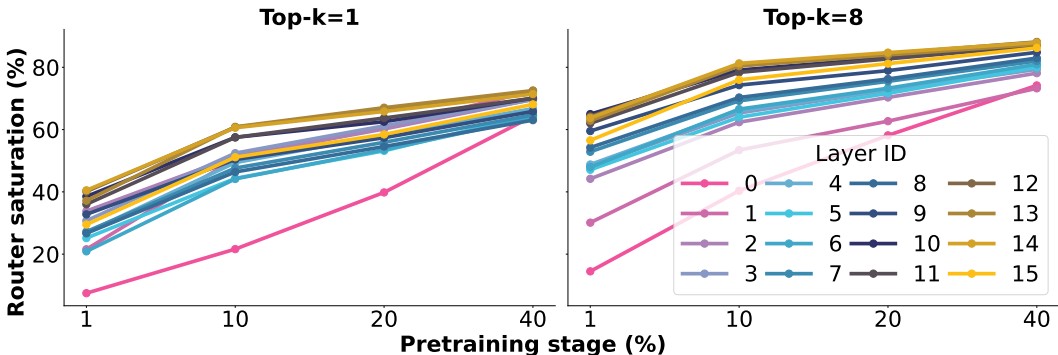

Figure 5: **Router saturation during pretraining measured on a random 0.5% of the C4 validation data.** We compute saturation by comparing the routing to the top-$k$ experts at four intermediate checkpoints (1, 10, 20, and 40% of pretraining) to the final pretraining checkpoint (Equation 5).

corresponds to whether the router weights are still learning which expert will process certain data. A value of 100% indicates that the router at the intermediate checkpoint will route to the same experts as the final checkpoint router. See §H.1 for the detailed formula used to calculate the value.

Figure 5 shows that, after 1% of pretraining, up to ~60% of routing to the top-8 activated experts has already saturated (right). Thus the model already uses the same 8 experts for given input data as it will at the end of pretraining. This early saturation aligns with prior work (Xue et al., 2024). At 40% of pretraining, saturation reaches up to ~80%. However, which top-1 expert has the highest routing probability saturates slower (left). We find that routing in later layers saturates earlier during pretraining. Layer 0 is an outlier saturating significantly more slowly than other layers. Dai et al. (2024) do not use an MoE in the first layer as they find that load balancing converges more slowly for the first layer. This is likely linked to our findings on saturation. Because routing in the first layer saturates slower, the experts that certain input data get routed to frequently change. These changes may lead to one expert suddenly getting significantly more data than others thereby impairing load balancing. We are excited about future work further investigating what happens in the first layer by building on our open release.

## 5.2 EXPERT CO-ACTIVATION

We define expert co-activation as the proportion of times two specific experts, $E_i$ and $E_j$, are simultaneously activated out of the total number of activations of one of them. A co-activation of 100% indicates that if $E_i$ is activated, $E_j$ is also always activated. A value of 0% indicates that the experts never co-occur. See §H.1 for the formula used to calculate the value.

Figure 6 shows there is no strong co-activation among experts in layer 7, with only few exceptions. This may indicate that there is little redundancy across different experts. Layers 7 and 15 (Figure H1) show similar co-activation patterns with several groups of 3 or 2 experts that tend to get activated together. We investigate tokens that activate these experts in §5.4. Further, in §H.2 (Figure H8), we investigate whether experts across layers, rather than within one layer, tend to process tokens together.

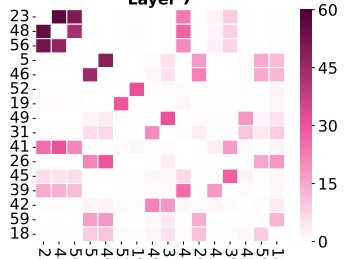

Figure 6: **Co-activation of OLMOE-1B-7B experts on a random 0.5% of the C4 validation data.** We display the 32 experts with the highest maximum co-activation score via their expert IDs on the x- and y-axis. See Figure H1 for Layer 0 and 15.

## 5.3 DOMAIN SPECIALIZATION

We define domain specialization as the specialization of expert $E_i$ to domain $D$, specifically the proportion of tokens from a particular domain $D$ that get routed to a particular expert $E_i$ (see §H.1 for the formula). A value of 100% indicates that all data from that domain is routed to $E_i$, whereas

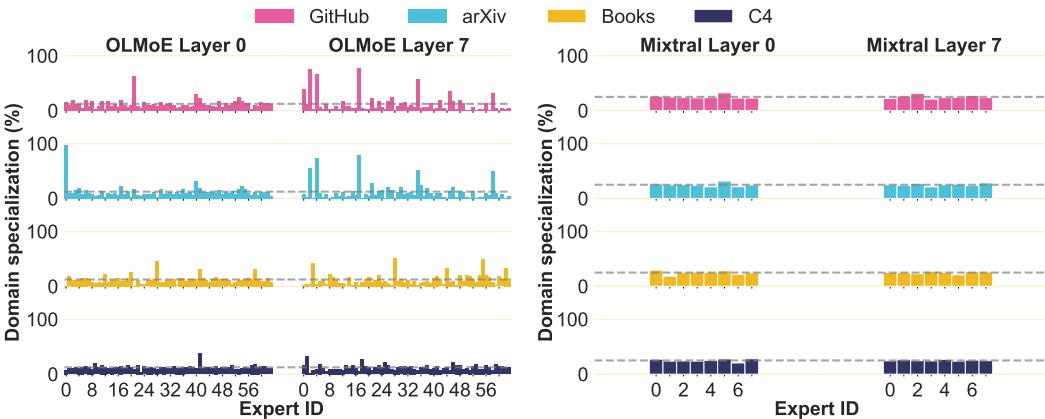

Figure 7: **Domain specialization of OLMOE-1B-7B (left) vs. Mixtral-8x7B (right).** We visualize how often tokens from different domains get routed to the 64 (**OLMOE**) or 8 (Mixtral) experts after pretraining. We consider tokens routed to any of the $k = 8$ (**OLMOE**) or $k = 2$ (Mixtral) active experts (Equation 7). Horizontal gray lines are random chance or uniform routing (8/64=12.5% per expert for **OLMOE-1B-7B** with 8 active out of 64 total experts per layer and 2/8=25% for Mixtral with 2 active out of 8 total experts per layer). See Figure H7 for $k = 1$ results.

0% indicates the expert is never used for that domain and can be removed from the model without affecting performance in that domain.

Figure 7 (left) shows many examples of experts that are activated significantly above or below random chance for *specific domains*. E.g., for arXiv, which has a very specific distribution with lots of scientific text, the first expert in layer 0 is nearly 100% specialized. This suggests that there is little redundancy in the knowledge of the experts in **OLMOE-1B-7B**, as they specialize in different kinds of data. GitHub and arXiv are often activated together in layer 7, which we explore further in §5.4. For *generic domains*, such as C4 (Raffel et al., 2023), which is a web crawl containing various kinds of data, expert activations in **OLMOE-1B-7B** are much more balanced. This highlights that the load balancing (§B.1.3) works as intended and the model makes proper use of all experts for generic data. Mixtral-8x7B (Jiang et al., 2024) in Figure 7 (right), however, exhibits little domain specialization across both *unique* and *generic domains*. Experts are activated close to the uniform routing baseline for all layers and domains. Thus, there may be more redundancy across experts in Mixtral, as they likely contain similar knowledge. We hypothesize that this is due to Mixtral being upcycled from Mistral (Cai, 2023). The initialization from a dense model may limit the amount of possible specialization in the experts as they all start from the same local optimum. This is likely why training from scratch eventually outperforms upcycling in our pretraining experiments (§B.1.2).

## 5.4 VOCABULARY SPECIALIZATION

Vocabulary specialization refers to how specialized a particular expert is on a token ID $x$ (also called a vocabulary element), defined as the proportion of tokens with a token ID $x$ that are routed to one particular expert $E_i$ out of all experts in that layer. We distinguish input and output variants of this specialization, where $x$ is either the input token ID or the next output token ID (either the ground-truth next token ID or the token ID predicted by the model). A value of 100% indicates that for all occurrences of that vocabulary element, input data is routed to $E_i$, whereas 0% indicates an expert that is fully irrelevant for that vocabulary element and can be effectively removed from the model without affecting performance whenever the token ID appears.

In Figure H2 we find that vocabulary specialization is higher in later layers, similar to how later layers saturate earlier (§5.1). Later layers also specialize more on predicted output token IDs rather than input token IDs, i.e., the routing is decided more by the token the model is about to predict rather than the original input token. This is intuitive as in earlier layers there is more uncertainty about which token the model will predict. At ~90%, expert 27 specializes the most, which we find in Table 4 to activate for many non-alphabetic tokens, such as Cyrillic and Devanagari letters. Expert 43 shows specialization on geographic terms in both input and output tokens. Experts 48 and 23 both focus on

Table 4: **Vocabulary specialization in the 7th layer of OLMoE-1B-7B.** We use $k = 1$ (Equation 8) and a random 0.5% of the C4 validation data excluding token IDs with <10 appearances.

| Expert ID | Input token IDs | Predicted output token IDs |
|---|---|---|
| 27 | © (100%) I· (100%) ³ (100%) i (100%) ǐ (100%) ◌̣ (100%) ◌̇ (100%) फ (100%) Ш (100%) ИН (100%) a (100%) | ľ (100%) § (100%) © (100%) ĵ (100%) ij (100%) O (100%) ∅ (100%) र (100%) फ (100%) ◌̣ (100%) ल (100%) |
| 43 | Armenian (100%) ijan (100%) enia (96%) Iraq (95%) Iranian (92%) Iran (92%) Saudi (90%) northern (90%) Lebanon (90%) Singapore (88%) Turkey (88%) | enia (90%) invasion (80%) Arabia (76%) irregular (66%) regions (64%) border (63%) Kong (61%) ians (61%) bases (60%) Republic (59%) Ireland (58%) |
| 4 | sq (89%) Main (70%) reversal (69%) YR (63%) GC (56%) Overall (50%) 79 (50%) main (50%) RE (46%) PCR (46%) | YR (90%) Character (88%) sq (77%) Os (76%) GHz (71%) fluence (60%) amycin (60%) pixels (56%) = (53%) |
| 48 | compared (42%) !) (41%) Then (41%) ', (40%) ), (35%) ", (35%) instead (33%) | except (60%) tennis (41%) Marks (40%) Dunn (33%) tears (30%) Arizona (30%) |
| 23 | .... (58%) Therefore (55%) So (46%) !!! (46%) And (44%) According (41%) ." (41%) !! (40%) ?" (38%) But (38%) | Đ (53%) Republican (50%) Jack (47%) THIS (40%) Democratic (40%) according (39%) So (38%) Step (33%) |
| 3 | grandmother (92%) brother (91%) Daisy (83%) daughter (78%) mum (75%) | hood (36%) mother (35%) inde (31%) boy (29%) girl (28%) married (27%) |

connector words, such as Then and Therefore . This is likely because they commonly process tokens together with a high co-activation of 60% in Figure 6. Based on our findings in §5.3 that for GitHub and arXiv often the same experts in layer 7 activate, we display one such expert (expert ID 4) in Table 4. It seems to specialize in measurements, such as sq , YR (year), and GHz . These are common terms in scientific papers corresponding to the arXiv domain and likely also in GitHub code for computations related to measurements. They are less likely to appear in books, which explains the low activation of expert ID 4 in layer 7 for book data in Figure 7. Expert 3 is among the three most active experts of layer 7 for book data in Figure 7 (fourth yellow bar for layer 7). This resonates when looking at its specialization on family terms in Table 4 (final row), which are far more common in books than scientific papers or code. Overall, domain specialization and vocabulary specialization are closely linked to one another, as domains are usually characterized by their distinct word distribution. In Figure H5, we link them more closely by comparing the extent of vocabulary specialization across domains and expert IDs. In Figure H3 and Figure H4 we also find that **OLMoE-1B-7B** exhibits stronger vocabulary specialization than Mixtral-8x7B.

## 6  CONCLUSION

We open-source **OLMoE-1B-7B** and **OLMoE-1B-7B-INSTRUCT** including model, data, code, and logs. At 1B active and 7B total parameters, our models yield state-of-the-art performance among models with a similar amount of active parameters even outperforming larger models including DeepSeekMoE-16B and Llama2-13B-Chat. We share ~20 training experiments yielding novel insights into Mixture-of-Experts. Further, we define and analyze new properties of MoEs showing that **OLMoE-1B-7B** exhibits early router saturation, weak expert co-activation, and some evidence of domain and vocabulary specialization. We intend our fully open release to serve as a basis for more research into MoEs given their critical importance (§1). We are excited about more iterations of **OLMoE** to close the gap between frontier models and fully open models.

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

## A RELATED WORK

**Advances in MoEs** Current LMs still largely follow the transformer architecture (Vaswani et al., 2023) with only few architectural changes that have been widely adopted, such as decoder-only training (Radford et al., 2019), SwiGLU activations (Shazeer, 2020; Dauphin et al., 2017), RoPE (Su et al., 2023), MQA/GQA (Shazeer, 2019; Ainslie et al., 2023) and RMSNorm (Zhang & Sennrich, 2019). Model sparsity via Mixture-of-Experts is one modification still under active exploration with some early adoption but most LMs, including Llama 3 (Dubey et al., 2024), still rely on a dense architecture. There has been a lot of progress in improving the sparsely-gated MoE layer since its introduction (Shazeer et al., 2017): New routing techniques (Lewis et al., 2021; Roller et al., 2021; Zuo et al., 2022; Gross et al., 2017; Jaszczur et al., 2021; Dua et al., 2021; Zhong et al., 2024; Wu et al., 2024b; Muqeeth et al., 2024), fine-grained expert segmentation (Dai et al., 2024; He, 2024), stability (Zoph et al., 2022) and efficiency (Lepikhin et al., 2020; Rajbhandari et al., 2022; Du et al., 2022; Zhou et al., 2024; Li et al., 2022; Sukhbaatar et al., 2024; Pan et al., 2024; Ren et al., 2023) improvements. In this work, we perform many experiments to provide insights into training Mixture-of-Experts LMs. Subsequently, we train **OLMOE-1B-7B** for 5T tokens. No prior MoE has been overtrained (Gadre et al., 2024) to this extent to our knowledge making **OLMOE-1B-7B** the best testbed to research performance saturation of MoEs vs. dense models. With **OLMOE** we hope to facilitate such and other research to help the field uncover whether MoEs should make it into all future LMs and with what precise configuration.

**Open LMs** A variety of model families have been proposed under varying degrees of openness commonly categorized based on whether model weights are available. **Closed-weight** models include GPT (Brown et al., 2020; OpenAI et al., 2023), Gemini (Team et al., 2023; 2024a), PaLM (Chowdhery et al., 2022; Anil et al., 2023), Reka (Team et al., 2024e), and **open-weight** ones include Llama (Touvron et al., 2023a;b; Dubey et al., 2024), Mistral (Jiang et al., 2023; 2024), Gemma (Team et al., 2024b;c), Falcon (Almazrouei et al., 2023; Penedo et al., 2023), MPT (Team, 2023), Qwen (Bai et al., 2023a; Yang et al., 2024a), GLM (GLM et al., 2024), Yi (AI et al., 2024), DeepSeek (DeepSeek-AI et al., 2024a;b; Dai et al., 2024), Nemotron (Parmar et al., 2024; Nvidia et al., 2024; Wang et al., 2024b), InternLM (Cai et al., 2024), Baichuan (Yang et al., 2023), Phi (Gunasekar et al., 2023; Li et al., 2023c; Abdin et al., 2024), StableLM (Bellagente et al., 2024), OPT (Zhang et al., 2022), Zamba (Glorioso et al., 2024). However, besides model weights, training data and code are key to enabling scientific research of these models (Longpre et al., 2023b; 2024) and distributing their benefits broadly (Bommasani et al., 2023). There have been few releases also including data and code in addition to model weights which we refer to as **"fully open-source"**: BLOOM (Workshop et al., 2023; Scao et al., 2022; Muennighoff et al., 2023c; Yong et al., 2023), GPT-NeoX (Black et al., 2022; 2021; Wang & Komatsuzaki, 2021), StarCoder (Li et al., 2023a; Lozhkov et al., 2024; Allal et al., 2023; Muennighoff et al., 2023a; Zhuo et al., 2024), Pythia (Biderman et al., 2023), OLMo (Groeneveld et al., 2024), LLM360 (Liu et al., 2023), Cerebras-GPT (Dey et al., 2023), DCLM (Li et al., 2024a), MAP-Neo (Zhang et al., 2024a), RWKV (Peng et al., 2023; 2024), and SmolLM (Allal et al., 2024). For Mixture-of-Experts only OpenMoE (Xue et al., 2024) aims to be fully open-source, however, its poor performance limits its usefulness. We release **OLMOE-1B-7B** as the first state-of-the-art Mixture-of-Experts LM that is fully open-source: model weights, data, code, and logs.

# B   ADDITIONAL EXPERIMENTS ON ALTERNATIVE DESIGN CHOICES

In this section, we present an extension of pretraining and adaptation experiments that have led to **OLMoE-1B-7B** (also see §4). We group them into experiments on settings specific to Mixture-of-Experts (§B.1), experiments on settings applicable to both dense LMs and MoEs (§B.2), and adaptation experiments (§B.3). In pretraining experiments, we often use MMLU Var, a version of MMLU (Hendrycks et al., 2021a) with varying few-shots and a different format that provides signal earlier during training. We describe our full evaluation setup in Appendix D and provide additional experiments in Appendix G. Each experiment links to a Weights & Biases report with more validation and downstream results, and the full configurations of the runs. To isolate the impact of changes and minimize confounders, we vary only one hyperparameter for each experiment. Nevertheless, due to the large number of hyperparameters, some results may change under different configurations and we cannot guarantee the correctness of each of our hyperparameter choices. Models are not comparable across different experiments, as we vary the base model to incorporate successful findings.

## B.1   MoE-SPECIFIC PRETRAINING SETTINGS

### B.1.1   SHARED EXPERTS

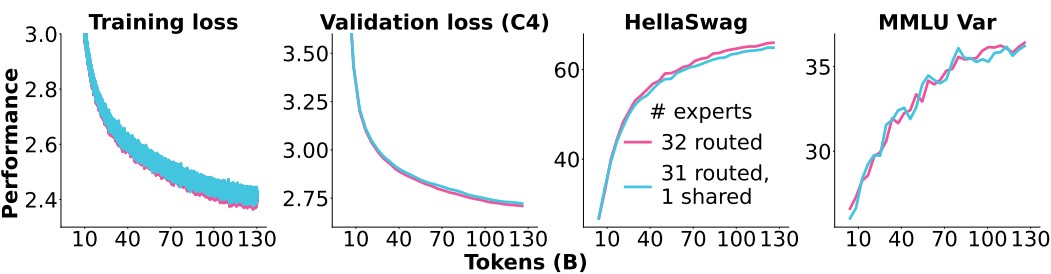

Figure B1: **Shared experts.** Both setups have the same number of active and total parameters and use the same number of FLOPs. 4 of the 32 routed experts are activated, while it is 3 for the 31 routed experts of the other model, as it has 1 always-active shared expert. More results, logs, and configurations: https://wandb.ai/ai2-llm/olmoe/reports/Plot-Expert-sharing--Vmlldzo4OTIyMjQz

Dai et al. (2024) propose training with a shared/fixed expert that is always used in addition to the routed experts. The intuition is to encourage the shared expert to learn common information and allow the other routed experts to learn more specialized knowledge. This should reduce redundancy among experts and thus lead to a better model as it can store more total information.

In Figure B1, we benchmark having a single shared and a single routed expert versus two routed experts. While both settings lead to similar performance, sharing an expert performs slightly worse. Sharing an expert removes flexibility from the model and thus goes against the findings in §4.2 suggesting that allowing for more expert combinations improves performance. Specifically, the two models in Figure B1 have $\binom{32}{4} = 35,960$ and $\binom{31}{3} = 4,495$ possible combinations per layer. Thus, removing one of the routed experts and turning it into a shared one eliminates almost 90% of possible combinations. This likely acts as a counterforce to the potential benefits of isolating common knowledge in a shared expert. Based on these results, we do not use shared experts in **OLMoE-1B-7B**, but we do think that there is merit to the idea of experts that are activated more often or even always. However, rather than enforcing this behavior via a shared expert, we believe that it should be learned by the model. This is difficult with current setups due to the necessity of a load balancing loss (§B.1.3) penalizing the model if tokens are not distributed equally among experts. Potential future work can explore removing the load balancing loss to allow for more flexible usage of experts.

### B.1.2   SPARSE UPCYCLING

Komatsuzaki et al. (2023) propose turning a dense model into a Mixture-of-Experts model via sparse upcycling: (1) The dense MLP is cloned for each desired expert to constitute MoE layers. (2) A newly

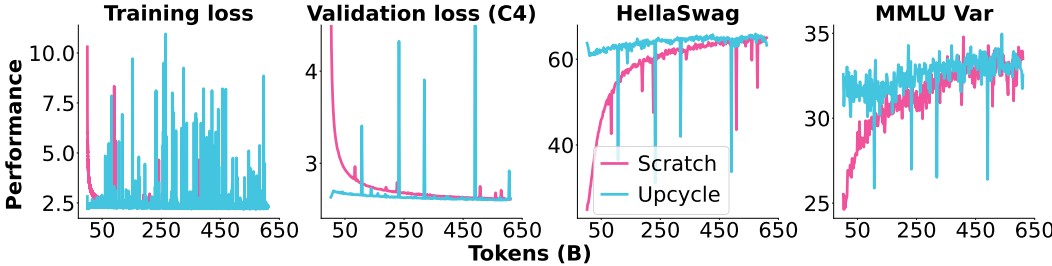

Figure B2: **Sparse upcycling.** We upcycle OLMo-1B (0724) at 2T tokens into an MoE with 8 total experts of which 2 are activated and train it for an additional 610 billion tokens. We compare it to a model trained from scratch for 610 billion tokens. Except for this difference, both models use the same config, which includes some suboptimal settings that contribute to the instability, such as no QK-Norm (§B.2.5) and no truncated normal init (§B.2.2). More results, logs, and configurations: https://wandb.ai/ai2-llm/olmoe/reports/ Plot-Scratch-vs-Upcycle--Vmlldzo4NDIyOTc4

initialized router is added in front of each MoE layer. (3) Pretraining continues with the new model so that the cloned MLPs can gradually specialize in different things and the router can be learned. They find that the upcycling approach maintains a performance advantage over a language model trained from scratch for up to 120% of the compute budget of the original dense checkpoint that the sparse model was upcycled from. For example, if sparsely upcycling a 1.3B parameter model at 2 trillion tokens then only at 2.4 trillion tokens should an MoE trained from scratch catch up with the upcycled model. That is, the sparsely upcycled model would have been trained for another 400 billion tokens, thereby saving the equivalent of up to 2T tokens of compute. Other works such as MiniCPM (Hu et al., 2024), Qwen2 (Yang et al., 2024a) and reportedly Mixtral (Cai, 2023; Jiang et al., 2024) have adopted sparse upcycling but only share limited information about their configuration.

In Figure B2, we compare sparse upcycling OLMo-1B (0724) (Groeneveld et al., 2024) with training an MoE from scratch. We find that after 500B tokens, an otherwise equivalent MoE trained from scratch already catches up with the upcycled model both on the metrics in Figure B2 and our additional metrics at https://wandb.ai/ai2-llm/olmoe/reports/ Plot-Scratch-vs-Upcycle--Vmlldzo4NDIyOTc4. At around 600B tokens, the MoE from scratch starts outperforming the upcycled MoE. Thus, it only requires 25% of the compute budget of the original dense model to catch up as opposed to the 120% reported in Komatsuzaki et al. (2023). However, they use expert choice routing and study encoder-decoder models (Raffel et al., 2023). Meanwhile, we use token choice routing (§4.3) and decoder-only models (§2). Further, we upcycle a model that has already been significantly overtrained (Gadre et al., 2024), i.e., a 1B model trained for 2T tokens. Its parameters are likely already in a very optimal range for a dense model, which may limit the amount of additional exploration possible after upcycling. This motivates us to experiment with adding noise to the upcycled weights outlined in Appendix G, but we do not find it to lead to better performance. A large disadvantage of upcycling is that the upcycled MoE is constrained by some hyperparameters of the dense model. Specifically, OLMo-1B (0724) was trained without QK-Norm and normal initialization, both of which hurt stability in our experiments (§B.2.5, §B.2.2). While it may be possible to simply add new QK-Norms and train them from scratch similar to the new router layer trained from scratch, it is impossible to change the initialization of the original dense model when upcycling it. Thus, as we want to change these hyperparameters and also train **OLMoE-1B-7B** for around 250% of the compute budget of the dense model (5T vs. 2T tokens), we do not use upcycling.

### B.1.3 LOAD BALANCING LOSS

Shazeer et al. (2017) propose the load balancing loss to penalize the model if it is unbalanced, i.e., if it routes all tokens to only a few experts. This is based on the observation that without such penalty, models tend to update only a select few experts in each layer (Eigen et al., 2014; Bengio et al., 2016). To compute the load balancing loss ($\mathcal{L}_{LB}$) we multiply the fraction of tokens $f_i$ routed to one expert $E_i$ with the total routing probability $P_i$ allocated to $E_i$ for one batch and sum it across the number of

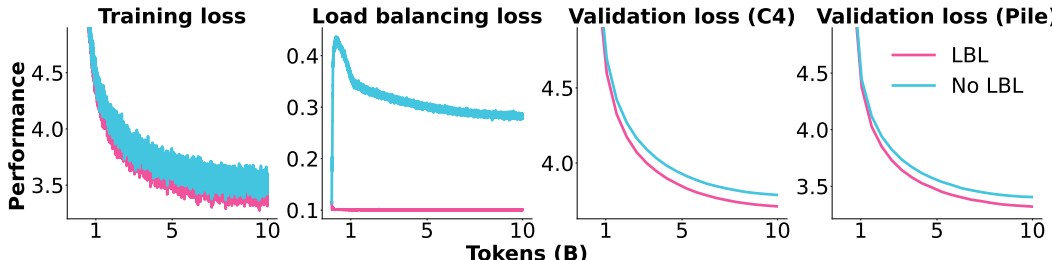

Figure B3: **Impact of applying a load balancing loss (LBL).** The training loss plot excludes the load balancing loss for both models. More results, logs, and configurations: `https://wandb.ai/ai2-llm/olmoe/reports/Plot-LBL-vs-No-LBL--Vmlldzo4OTkyNDg4`

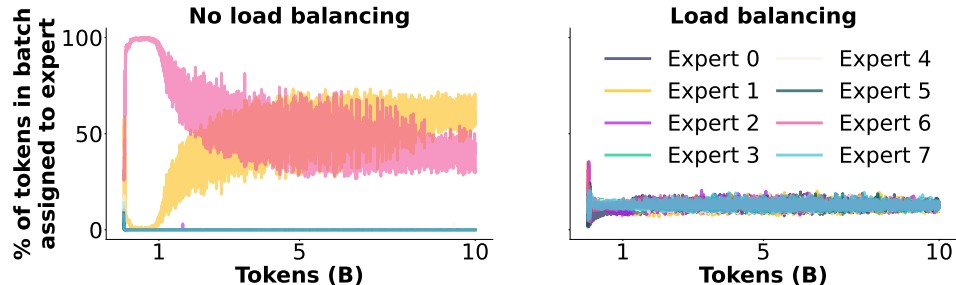

Figure B4: **Expert assignment during training when using or not using a load balancing loss for the first MoE layer.** More results, logs, and configurations: `https://wandb.ai/ai2-llm/olmoe/reports/Plot-LBL-vs-No-LBL--Vmlldzo4OTkyNDg4`

experts $N_E$:

$$\mathcal{L}_{LB} = N_E \cdot \sum_{i=1}^{N_E} f_i \cdot P_i \tag{3}$$

The loss is further scaled by $N_E$ and a loss weight $\alpha$ (see Equation 2), which is an optional weight to determine the magnitude of the loss commonly set to 0.01 (Zoph et al., 2022; Xue et al., 2024). We do not experiment with changing the weight of 0.01.

In Figure B3 we investigate the performance impact of using the auxiliary load balancing loss. We find that across training loss and validation losses, using the load balancing loss leads to better performance even after only a few billion tokens. We still measure the load balancing loss even when it is not used ("No LBL") and find that while it spikes initially, it slowly decreases over the next few billion tokens. This behavior is also visible in Figure B4 (left), where initially all tokens in the first layer are assigned to the 6th expert (pink). Eventually, the model also starts assigning some tokens to the 1st expert (yellow). However, all other experts remain largely flat and are thus "dead weights" that take up GPU memory but are not used. Given these results, we use the auxiliary load balancing loss with a weight of 0.01 following prior work (Shazeer et al., 2017; Shen et al., 2024). However, getting rid of the load balancing loss is an important direction for future research as it constrains the flexibility of the model by forcing it to use all experts approximately equally. This could prevent the experts from specializing in certain data domains and may be a reason prior work has failed to find strong evidence of expert specialization (Jiang et al., 2024; Zoph et al., 2022).

### B.1.4 ROUTER Z-LOSS

Zoph et al. (2022) propose the router z-loss to improve both the stability and quality of MoE models. This auxiliary loss penalizes large logits coming into the gating network. Such large logits can lead to numeric overflows in the large matrix multiplications happening in the MoE layer. It is computed

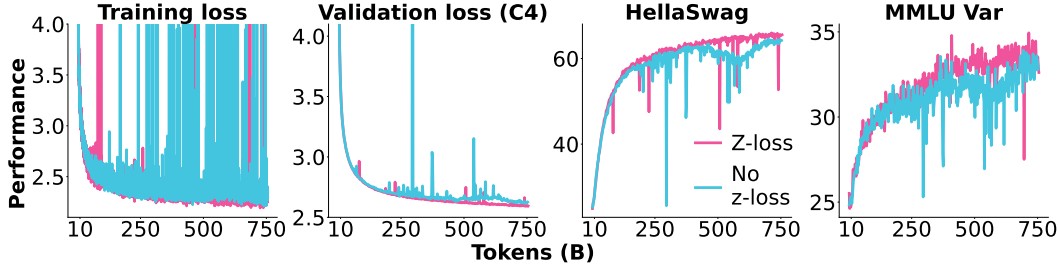

Figure B5: **Router z-loss.** We compare adding router z-loss with a loss weight of 0.001 versus no additional z-loss. More results, logs, and configurations: `https://wandb.ai/ai2-llm/olmoe/reports/Plot-Zloss-vs-none--Vmlldzo4NDM4NjUz`

by exponentiating the logits $x_j$ right before the router layer summed across the number of experts $N_E$ and averaged across the batch $B$, thereby making larger logits lead to a larger loss:

$$\mathcal{L}_{RZ}(x) = \frac{1}{B} \cdot \sum_{i=1}^{B} \left( \log \sum_{j=1}^{N_E} \exp(x_j^{(i)}) \right)^2 \tag{4}$$

The loss is further multiplied with an optional loss weight, $\beta$ (see Equation 2), to determine the magnitude of the loss commonly set to 0.001 (Zoph et al., 2022; Shen et al., 2024). We do not experiment with changing the weight of 0.001.

In Figure B5, we confirm that across training loss, validation loss, and downstream performance adding the router z-loss improves stability (less spikes) and quality (lower loss and higher downstream performance). Thus, despite it reducing throughput by ~2% we use the router z-loss for **OLMoE-1B-7B** with a weight of 0.001 as in Zoph et al. (2022).

## B.2 GENERAL PRETRAINING SETTINGS

### B.2.1 DATASET EXPERIMENTS

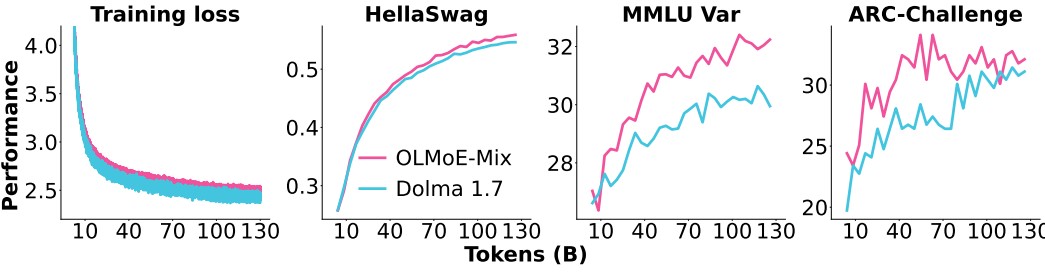

Figure B6: **OLMoE-MIX vs. Dolma 1.7.** We compare our data mix described in §2 with Dolma 1.7 used to train prior OLMo models. Lower training loss does not mean that one dataset is better, but rather suggests which dataset is easier for the model to learn. More results, logs, and configurations: `https://wandb.ai/ai2-llm/olmoe/reports/Plot-Dolma-1-7-vs-Dolma-OLMoE--Vmlldzo4OTIxNTg5`

Li et al. (2024a) release the DCLM-Baseline dataset and establish that it leads to better language models than Dolma 1.7 and other datasets as measured on common benchmarks like MMLU (Hendrycks et al., 2021a). This motivates us to mix their DCLM dataset with some components from Dolma 1.7 that we deem to be high-quality; see §2. In Figure B6, we compare our mix, **OLMoE-MIX**, with Dolma 1.7 in a controlled setup. We find that **OLMoE-MIX** leads to clear gains on all three downstream metrics, especially MMLU. DCLM-Baseline has been created through a series of dataset ablations targeting MMLU and other downstream metrics, which explains these results. We also compare

adding Reddit and FLAN to our mix as detailed in Appendix G, but do not find consistent performance gains. We do not have a strong intuition for why adding these datasets does not help and a more automatic approach to dataset mixing may be desirable for future iterations (Liu et al., 2024a; Albalak et al., 2024). We pretrain using our mix of DCLM-Baseline and Dolma 1.7 dubbed **OLMoE-Mix**.

### B.2.2 INITIALIZATION

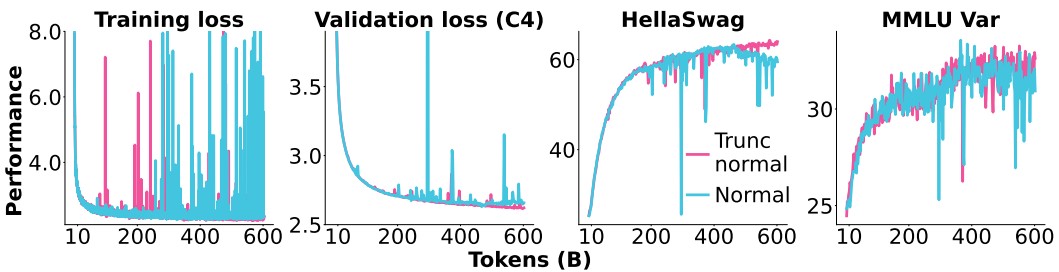

Figure B7: **Initialization.** We compare a normal initialization with a standard deviation (std) of 0.02 with a truncated normal initialization with a maximum (minimum) cut-off of 0.06 (–0.06) corresponding to three stds (3×0.02). More results, logs, and configurations: `https://wandb.ai/ai2-llm/olmoe/reports/Plot-Init--Vmlldzo4NDIzMzM5`

Few prior works on Mixture-of-Experts share their initialization strategy. Even the most open MoEs prior to this work, JetMoE (Shen et al., 2024) and OpenMoE (Xue et al., 2024), do not mention their initialization scheme. For DeepSeekMoE (Dai et al., 2024) and DeepSeekV2 (DeepSeek-AI et al., 2024b), the authors share that they use a normal initialization with a standard deviation (std) of 0.006. For dense language models, a normal initialization with an std of 0.02 has been commonly used as popularized by Shoeybi et al. (2020).

In Figure B7, we find a truncated normal initialization leads to more stable training and better performance than a regular normal initialization. The difference between the two initializations only becomes clear at around 450 billion tokens, where the model with the normal initialization starts to diverge. This is despite both models using the same configuration except for the difference in weight initialization. Having to train for hundreds of billions of tokens until an experiment provides a clear signal is one of the key challenges of pretraining ablations. We use the truncated normal initialization for **OLMoE-1B-7B**.

### B.2.3 RMSNORM

OLMo (Groeneveld et al., 2024) uses non-parametric layer normalization (Ba et al., 2016), mainly as it is significantly faster than the commonly used RMSNorm (Zhang & Sennrich, 2019; Mehta et al., 2024). This is an unusual choice as most LMs use RMSNorm, such as the Llama (Touvron et al., 2023a;b; Dubey et al., 2024), Gemma (Team et al., 2024b;c), and Qwen (Bai et al., 2023a; Yang et al., 2024a) model families.

In Figure B8, we observe that replacing the non-parametric layer normalization in OLMo with a parametric RMSNorm leads to better performance. This is likely because the non-parametric layer normalization leads to a large number of spikes in the gradients as seen in Figure B10. We clip gradients at 1.0, which prevents these spikes from leading to very large and potentially disruptive parameter updates. However, the clipped gradients may still harm the performance of the model as they are no longer the true gradients. Thus, despite RMSNorm lowering our training throughput by 15%, we train our final model with RMSNorm. We include the RMSNorm parameters in weight decay

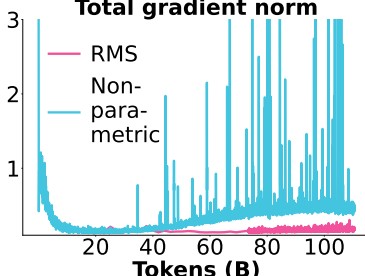

Figure B10: **Total norm of the gradients when training with RMS or non-parametric normalization.** We increase the logging interval of the RMS run at 75B tokens, hence its change in thickness.

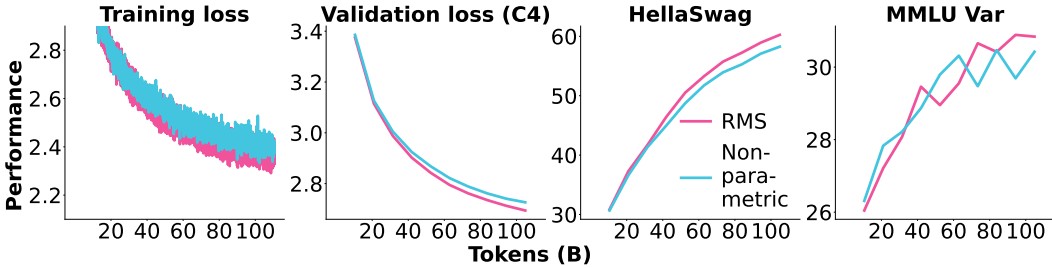

Figure B8: **Non-parametric layer normalization vs. RMSNorm.** More results, logs, and configurations: https://wandb.ai/ai2-llm/olmoe/reports/Plot-LN--Vmlldzo4NDQyMTAz

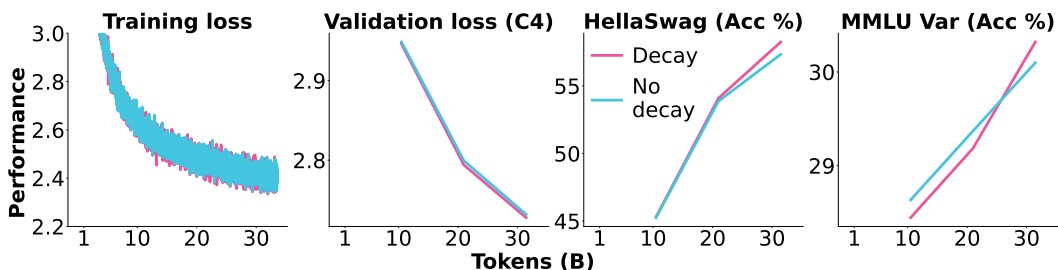

Figure B9: **Decaying the RMSNorm parameters.** More results, logs, and configurations: https://wandb.ai/ai2-llm/olmoe/reports/Plot-Decay-LN--Vmlldzo4NDQ1NDYy

as we find that it performs slightly better (Figure B9) even though it is common practice to exclude them.[4]

### B.2.4 DECAYING EMBEDDING PARAMETERS

Similar to the RMSNorm parameters (§B.2.3), embedding parameters are commonly excluded from weight decay.[5] In Figure B11 we find that whether or not they are decayed has only a minor impact on performance, with decaying being slightly better. Thus for simplicity, we weight decay all parameters in **OLMOE-1B-7B** including embedding and RMSNorm.

### B.2.5 QK-NORM

Some works have reported stability improvements from adding layer normalization after the query and key projections ("QK-Norm") (Team, 2024a; Mehta et al., 2024; Dehghani et al., 2023). QK-Norm can prevent the subsequent attention operation from leading to very large logits that may lead to numeric overflows and destabilize the network, especially when training in low precision. Like layer normalization at other places in the model, the QK-Norm could be non-parametric or use the parametric RMSNorm (§B.2.3).

In Figure B12, we compare using QK-Norm with no normalization after the query and key projections. We find that QK-Norm leads to some stability and performance improvements. We perform this experiment with non-parametric layer normalization as used in OLMo (Groeneveld et al., 2024), while we used parametric RMS layer normalization (Zhang & Sennrich, 2019) for **OLMOE-1B-7B** (§B.2.3). To ensure the benefit of QK-Norm is not an artifact of comparing with non-parametric layer normalization, we run another experiment with RMS layer normalization and

---

[4]https://github.com/karpathy/minGPT/pull/24#issuecomment-679316025
[5]https://github.com/karpathy/minGPT/pull/24#issuecomment-679316025

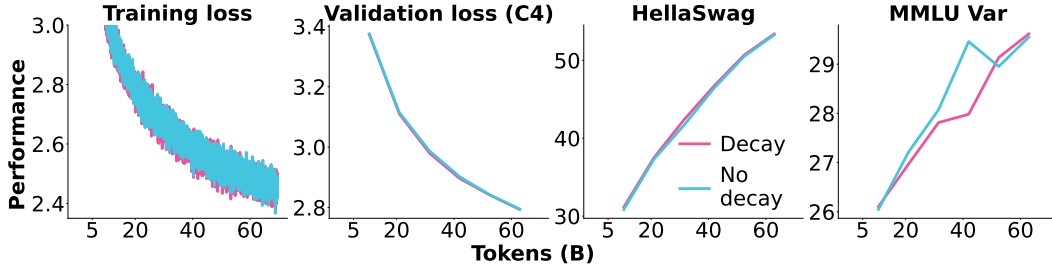

Figure B11: **Decaying the embedding parameters.** More results, logs, and configurations: `https://api.wandb.ai/links/ai2-llm/3h22onp5`

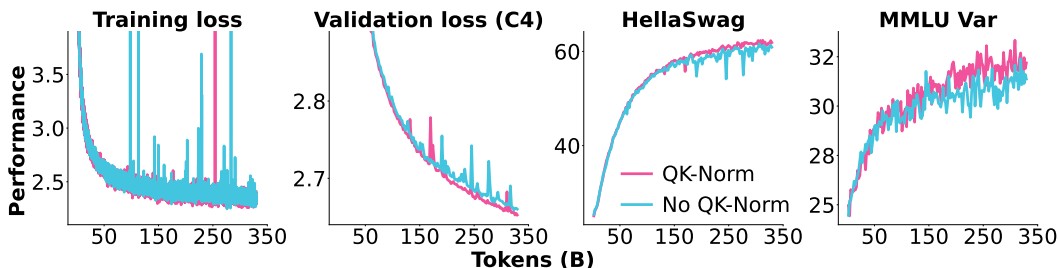

Figure B12: **Query-Key layer normalization (QK-Norm).** Both models use non-parametric layer normalization. QK-Norm corresponds to additional layer normalization of the query and key projections. More results, logs, and configurations: `https://wandb.ai/ai2-llm/olmoe/reports/Plot-QKNorm-vs-none--Vmlldzo4NDIzMzE2`

still find QK-Norm to lead to slightly better training loss and to prevent a large grad norm spike.[6] Thus, we use QK-Norm for **OLMoE-1B-7B** despite it reducing throughput by almost 10%.

### B.2.6    ADAMW EPSILON

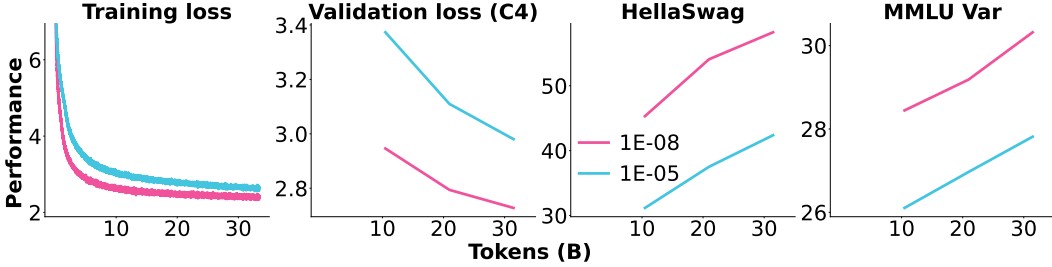

Figure B13: **AdamW epsilon.** More results, logs, and configurations: `https://wandb.ai/ai2-llm/olmoe/reports/Plot-AdamW-eps--Vmlldzo4NDc5MDg0`

Groeneveld et al. (2024) use an epsilon ("eps") value of 1E-05 in the AdamW optimizer for training OLMo. A larger eps value leads to smaller steps of the optimizer but can be more stable (Kingma & Ba, 2017).

In Figure B13, we find that decreasing eps to the recommended default of 1E-08 (Kingma & Ba, 2017) significantly improves performance while the run remains stable. Thus, we set eps to 1E-08 for our final run.

---

[6] `https://wandb.ai/ai2-llm/olmoe/reports/Plot-QKNorm-revisited--Vmlldzo4NTc2NTIz`

## B.3 ADAPTATION SETTINGS

Table B1: **Adaptation experiments of OLMOE-1B-7B.** We compare using the pretrained checkpoint prior to annealing for adaptation, using the checkpoint after the additional 100B tokens of annealing, and using the checkpoint after the additional 100B tokens of annealing and with load balancing loss (§B.1.3) during adaptation. We apply DPO/KTO to the respective SFT model.

| Task (→) | MMLU | GSM8k | BBH | Human-Eval | Alpaca-Eval 1.0 | XSTest | IFEval | Avg |
|---|---|---|---|---|---|---|---|---|
| Setup (→) | 0-shot | 8-shot CoT | 0-shot | 0-shot | 0-shot | 0-shot | 0-shot | 0-shot |
| Metric (→) | EM | EM | EM | Pass@10 | %win | F1 | Loose Acc | |
| **OLMOE-1B-7B** w/o annealing | 49.0 | 2.0 | 31.5 | 18.9 | - | 62.1 | 18.5 | - |
| +SFT | 50.2 | 43.0 | 35.6 | 55.5 | 68.9 | 83.8 | 39.7 | 53.8 |
| +DPO | 50.9 | 36.0 | 35.8 | **58.8** | 81.7 | 83.2 | 47.9 | 56.3 |
| **OLMOE-1B-7B** | 49.8 | 3.0 | 33.6 | 22.4 | - | 59.7 | 16.6 | - |
| +SFT | 51.4 | 40.5 | 38.0 | 51.6 | 69.2 | 84.1 | 43.3 | 54.0 |
| +DPO | **51.9** | **45.5** | 37.0 | 54.8 | **84.0** | 82.6 | **48.1** | **57.7** |
| +KTO | 51.2 | **45.5** | 34.1 | 57.1 | 81.6 | **86.6** | 47.5 | **57.7** |
| +SFT (load balancing) | 50.9 | 36.5 | 35.7 | 52.4 | 66.9 | 84.8 | 42.3 | 52.8 |
| +DPO (load balancing) | 51.1 | 42.5 | **39.3** | 55.6 | 82.9 | 82.1 | 46.0 | 57.1 |

We experiment with small design choices for adaptation using our evaluation setup described in Appendix D. **(1) Auxiliary losses:** Zoph et al. (2022) find that using the auxiliary load balancing loss (§B.1.3) during regular finetuning leads to small performance gains. For instruction tuning, however, Shen et al. (2023a) do not find conclusive evidence in favor of using the load balancing or router z-loss with only small differences in performance, both in support of and against the auxiliary losses. In Table B1 we display experiments with the load balancing loss during adaptation and find that not using it leads to better performance (54.0 vs. 52.8 after instruction tuning (SFT) and 57.7 vs. 57.1 after

Table B2: **Load balancing loss (Equation 3) over a subset of the respective corpora prior to scaling with the load balancing loss weight** $\alpha$. While we use load balancing loss during pretraining, we do not use it during SFT.

| Data (↓) | OLMOE-1B-7B | |
|---|---|---|
| | After pretraining | After SFT |
| Wikipedia | 8.331 | 8.367 |
| C4 | 8.073 | 8.076 |
| SFT data | 8.249 | 8.250 |

preference tuning (DPO)). One potential problem of deactivating the load balancing loss is that it may harm balance among experts and turn some into dead weights as observed during pretraining in §B.1.3. However, when measuring the load balancing loss in Table B2 on our SFT data (§2), we find that the loss only increases by around 0.01% after SFT (8.250 vs. 8.249). This is likely because which experts certain tokens get routed to is determined early during pretraining, as we find later in the analysis section (§5.1). We also visualize the activation patterns of experts of the model after pretraining, and the models after SFT and DPO trained without load balancing in §H.2 (Figure H6) finding that the distribution remains around the same. Thus, as our models adapted without load balancing perform better and we find it not to impact routing substantially, we do not use load balancing during adaptation .

**(2) Annealing checkpoint:** We also experiment with using the checkpoint pre-annealing (§2) for adaptation and find the checkpoint post-annealing leads to better performance (53.8 vs. 54.0 after SFT and 56.3 vs 57.7 after DPO), thus we use the post-annealing checkpoint. **(3) Preference algorithm:** Since the release of DPO (Direct Preference Optimization) (Rafailov et al., 2023), a variety of preference algorithms have been proposed (Ethayarajh et al., 2024; Hong et al., 2024; Meng et al., 2024). We experiment with KTO (Ethayarajh et al., 2024) and find that it matches DPO in Table B1 for our setup (Appendix C). While we release both models, we use DPO for our final

**OLMOE-1B-7B-INSTRUCT** model, as it scores higher on AlpacaEval, which has a smaller chance of data contamination than our other benchmarks (Xu et al., 2024).

## C    TRAINING CONFIGURATION

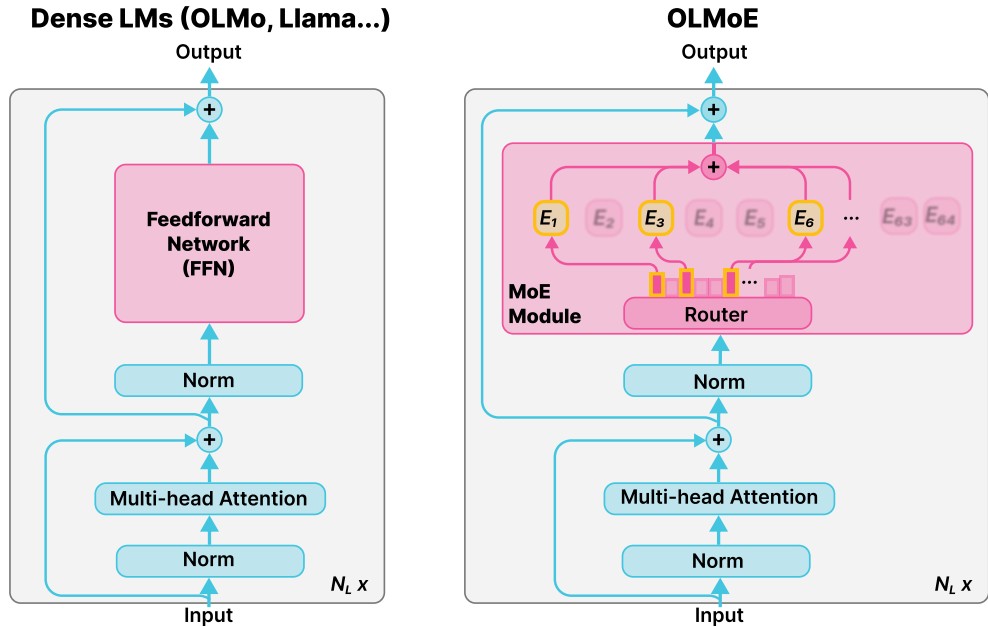

Figure C1: **Comparison of the architecture of dense LMs and MoE models like OLMOE.** The figure excludes some details, e.g., **OLMOE-1B-7B** also uses QK-Norm (§B.2.5).

Table C1: **Composition of the pretraining data for OLMOE-1B-7B**. StarCoder (Li et al., 2023a; Kocetkov et al., 2022), peS2o (Soldaini & Lo, 2023), and Wiki come from Dolma 1.7 (Soldaini et al., 2024). arXiv from Red-Pajama (Computer, 2023), OpenWebMath (Paster et al., 2023) and Algebraic Stack from ProofPile II (Azerbayev et al., 2023). Links to our data are in Appendix I.

| Source | Doc Type | GPT-NeoX tokens *(billions)* | Words *(billions)* | UTF-8 bytes *(GB)* | Documents *(millions)* |
|---|---|---|---|---|---|
| DCLM-Baseline | web pages | 3,860 | 3,380 | 16,700 | 2,950 |
| StarCoder | code | 101 | 63.9 | 325 | 78.7 |
| peS2o | STEM papers | 57.2 | 51.3 | 268 | 38.8 |
| arXiv | STEM papers | 21.1 | 23.5 | 88.8 | 1.55 |
| OpenWebMath | math web pages | 12.7 | 10.2 | 42.4 | 2.91 |
| Algebraic Stack | math proofs code | 12.6 | 9.6 | 39.3 | 2.83 |
| English Wikipedia & Wikibooks | encyclopedic | 3.69 | 3.16 | 16.2 | 6.17 |
| **Total** | | **4,060** | **3,530** | **17,400** | **3,080** |

**Pretraining**    We display the pretraining hyperparameter configuration of **OLMOE-1B-7B** in Appendix C comparing with other relevant models. We follow Groeneveld et al. (2024) using the AdamW optimizer (Loshchilov & Hutter, 2019) with ZeRO (Rajbhandari et al., 2020) via PyTorch FSDP (Zhao et al., 2023) and mixed-precision training (Micikevicius et al., 2018). Our main model settings differing from Groeneveld et al. (2024) are: **(1) MoE-related changes: OLMOE-1B-7B** is a sparsely activated decoder-only transformer (Vaswani et al., 2023) using dropless Mixture-of-Experts (Gale et al., 2022). Unlike most prior MoEs, we use a high granularity (Dai et al., 2024;

Table C2: **Adaptation training data for OLMOE-1B-7B.** We mix Tulu 2 (Ivison et al., 2023), No Robots (Rajani et al., 2023), CodeFeedback (Zheng et al., 2024), MetaMathQA (Yu et al., 2024) and Daring Anteater (Wang et al., 2024b) for SFT and use a filtered UltraFeedback (Cui et al., 2023; Lin et al., 2022) for preference tuning. Links to our data are in Appendix I.

| Source | Domain | Samples |
|---|---|---|
| *Instruction Tuning* | | |
| Tulu 2 SFT Mix | Various | 326,154 |
| No Robots | Various | 9,500 |
| CodeFeedback-Filtered-Instruction | Coding | 156,526 |
| MetaMathQA | Math | 98,750 |
| Advanced (non-chat) subset of Daring Anteater | Various | 17,082 |
| *Preference Tuning (DPO (Rafailov et al., 2023))* | | |
| UltraFeedback binarized and filtered for TruthfulQA contamination | Various | 60,800 |

Krajewski et al., 2024) with 64 small experts with an FFN dimension of just 1,024 rather than a few large experts. We further use two auxiliary losses: router z-loss (Zoph et al., 2022) and load balancing loss (Shazeer et al., 2017). **(2) Stability improvements:** (a) We use a truncated normal initialization with a standard deviation of 0.02 and a minimum (maximum) cut-off of -0.06 (0.06) corresponding to three standard deviations. (b) We use QK normalization (Team, 2024a; Mehta et al., 2024; Dehghani et al., 2023). (c) We use RMSNorm (Zhang & Sennrich, 2019) instead of the non-parametric LayerNorm used in Groeneveld et al. (2024). **(3) Performance improvements:** Besides some of the stability improvements which also impact performance, we also reduce the AdamW epsilon to 1.0E-08 from the 1.0E-05 used in Groeneveld et al. (2024) to speed up convergence. Finally, we train **OLMOE-1B-7B** for significantly longer than all prior OLMo models amounting to 5T tokens and thus more than one epoch (1.3) following Muennighoff et al. (2023b). We shuffle the pretraining dataset before starting the second epoch. To all data sources (Table C1), we apply a filter that removes all documents with a sequence of 32 or more repeated n-grams, where an n-gram is any span of 1 to 13 tokens. For the StarCoder subset, we also remove any document from a repository with fewer than 2 stars on GitHub, whose most frequent word constitutes over 30% of the document, or whose top-2 most frequent words constitute over 50% of the document. We shuffle all samples randomly at the beginning of each epoch and train for a total of 5.133T tokens. During our annealing phase (final 100B tokens), we reshuffle the entire dataset and then linearly decay the learning rate from 5.0E-04 to 0, following prior work (Groeneveld et al., 2024; Li et al., 2024a).

**Adaptation** For finetuning we use Open Instruct (Wang et al., 2023; Ivison et al., 2023).[7] We filter all SFT samples to a length of fewer than 4096 tokens to match the sequence length of the model. Following Muennighoff et al. (2024), we aggregate loss at the token level during SFT to improve performance on long generative tasks, such as AlpacaEval. We finetune in BF16 with a global batch size of 128 (4 H100 nodes with 8 GPUs each, a per device batch size of 2, and 2 gradient accumulation steps). We train for 2 epochs with a constant learning rate of 2.0E-5. For DPO (Rafailov et al., 2023), we reduce the global batch size to 32 (4 H100 nodes with 8 GPUs each and a per device batch size of 1). We train for 3 epochs with a learning rate of 5.0E-7 and a DPO beta of 0.1. Our adapted models are built on top of our annealed checkpoint, and we include the load balancing loss during both SFT and DPO based on our experiments in §B.3. Our preference tuning recipe is heavily optimized for DPO based on extensive experiments by Ivison et al. (2023), thus for KTO (Ethayarajh et al., 2024) we experiment with a few settings in Appendix G. Our final KTO adaptation uses the same hyperparameters as DPO, except that we use the RMSProp optimizer instead of Adam, which we use for SFT and DPO, and that we reduce the training duration to 1.3 epochs (5,000 steps) for KTO instead of the 3 epochs used for DPO.

**Hardware** We pretrain **OLMOE-1B-7B** on 256 H100 GPUs for approximately 10 days with NV-link interconnect across GPUs and InfiniBand interconnect across nodes. We also use H100 GPUs for all our experiments but some use a cluster with GCP TCPx interconnect across nodes

---

[7]Code: https://github.com/allenai/open-instruct

Table C3: **Pretraining hyperparameters of OLMOE-1B-7B and comparable models trained from scratch.** We highlight rows where **OLMOE-1B-7B** differs from OLMo-1B. Active params include vocab params. "?" = undisclosed settings, FFN = feed-forward network, Attn = Attention, LR = learning rate, WSD = Weight-Stable-Decay (Hu et al., 2024), LBL = load balancing loss, Inv Sq Root = Inverse Square Root decay (Shazeer & Stern, 2018), trunc = truncation, std = standard deviation, "varies" = stds that are layer or weight-dependent.

| | OLMOE-1B-7B | JetMoE | OpenMoE | OLMo-1B (0724) |
|---|---|---|---|---|
| Dimension | 2,048 | 2,048 | 2,048 | 2,048 |
| Activation | SwiGLU | SwiGLU | SwiGLU | SwiGLU |
| FFN dimension | 1,024 | 5,632 | 8,192 | 8,192 |
| Vocab size | 50,304 | 32,000 | 256,384 | 50,304 |
| Attn heads | 16 | 16 | 24 | 16 |
| Num layers | 16 | 24 | 32 | 16 |
| Layer norm type | RMSNorm | RMSNorm | RMSNorm | non-parametric |
| Layer norm eps | 1.0E-05 | 1.0E-05 | 1.0E-06 | 1.0E-05 |
| QK-Norm | yes | no | no | no |
| Pos emb. | RoPE | RoPE | RoPE | RoPE |
| RoPE $\theta$ | 10,000 | 10,000 | 10,000 | 10,000 |
| Attention variant | full | MoA | full | full |
| Biases | - | MLP & Attn | - | - |
| Weight tying | no | yes | no | no |
| Init dist | trunc normal | ? | ? | normal |
| Init std | 0.02 | 0.02 | varies | varies |
| Init trunc | 3×std | - | - | - |
| MoE layers | Every | Every | Every 6th | - |
| MoE layer type | dMoE | dMoE | ST-MoE | - |
| # Experts | 64 | 8 | 32 | 1 |
| # Activated | 8 | 2 | 2 | 1 |
| # Vocab params | 103M | 66M | 525M | 103M |
| # Active params | 1.3B | 2.2B | 2.6B | 1.3B |
| # Total params | 6.9B | 8.5B | 8.7B | 1.3B |
| Sequence length | 4,096 | 4,096 | 2,048 | 4,096 |
| Batch size (samples) | 1,024 | 1,024 | 2,048 | 512 |
| Batch size (tokens) | ~4M | ~4M | ~4M | ~2M |
| warmup steps | 2,500 | 2,500 | 10,000 | 2,000 |
| peak LR | 4.0E-04 | 5.0E-04 | 0.01 | 4.0E-04 |
| minimum LR | 4.0E-05 | 5.0E-05 | - | 4.0E-05 |
| optimizer | AdamW | AdamW | Adafactor | AdamW |
| weight decay | 0.1 | 0.1 | 0.0 | 0.1 |
| beta1 | 0.9 | ? | 0.9 | 0.9 |
| beta2 | 0.95 | ? | - | 0.95 |
| AdamW epsilon | 1.0E-08 | ? | - | 1.0E-05 |
| LR schedule | cosine | WSD | Inv Sq Root | cosine |
| gradient clipping | global 1.0 | global 1.0 | global 1.0 | global 1.0 |
| gradient reduce dtype | FP32 | ? | ? | FP32 |
| optimizer state dtype | FP32 | ? | ? | FP32 |
| LBL weight | 0.01 | 0.01 | 0.01 | - |
| Router z-loss weight | 0.001 | 0.001 | 0.0001 | - |
| Pretraining tokens | 5,033B | 1,000B | 1,100B | 2,000B |
| Annealing tokens | 100B | 250B | - | 50B |
| Annealing schedule | linear | - | - | linear |
| Annealing min LR | 0 | - | - | 0 |

instead. For adaptation, we use 32 H100 GPUs for 33 hours to instruction tune and for another 14 hours to preference tune via DPO. For KTO adaptation we use 8 H100 GPUs for 30 hours instead.

# D   EVALUATION SETUP

Table D1: **Summary of downstream evaluation during and after pretraining (OLMES).** ARC-C and ARC-E refer to ARC-Challenge and -Easy (Clark et al., 2018), CSQA=CommonsenseQA (Talmor et al., 2019), OBQA=OpenBookQA (Mihaylov et al., 2018), other benchmarks are named as in their original works (Clark et al., 2019; Gordon et al., 2012; Zellers et al., 2019; Hendrycks et al., 2021a; Bisk et al., 2019; Welbl et al., 2017; Sap et al., 2019; Sakaguchi et al., 2019). CF=Completion/Cloze formulation, MCF=Multiple-choice formulation, pmi=pointwise-mutual-information, Var=variants referring to the use of few-shots varying from 0-5.

| | During pretraining | | | | After pretraining (OLMES) | | | |
|---|---|---|---|---|---|---|---|---|
| Dataset ($\downarrow$) | Format | Shot | Norm | Split | Format | Shot | CF Norm | Split |
| ARC-C | CF | 0 | token | val | max(MCF,CF) | 5 | pmi | test |
| ARC-E | CF | 0 | none | val | max(MCF,CF) | 5 | character | test |
| BoolQ | CF | 0 | none | val | max(MCF,CF) | 5 | none | val |
| COPA | CF | 0 | none | val | - | - | - | - |
| CSQA | CF | 0 | token | val | max(MCF,CF) | 5 | pmi | val |
| HellaSwag | CF | 0 | token | val | max(MCF,CF) | 5 | character | val |
| MMLU | MCF | 5 | none | val | max(MCF,CF) | 5 | character | test |
| MMLU Var | CF | 0-5 | token | val | - | - | - | - |
| OBQA | CF | 0 | token | val | max(MCF,CF) | 5 | pmi | test |
| PIQA | CF | 0 | token | val | max(MCF,CF) | 5 | character | val |
| SciQ | CF | 0 | none | val | - | - | - | - |
| SocialIQA | CF | 0 | token | val | max(MCF,CF) | 5 | character | val |
| Winogrande | CF | 0 | none | val | max(MCF,CF) | 5 | none | val |

**During pretraining**   We evaluate using a similar in-loop evaluation setup as Groeneveld et al. (2024), with the addition of more tasks such as CommonsenseQA, PIQA, and different implementations of MMLU. Following Groeneveld et al. (2024), for the majority of the tasks, we perform 0-shot evaluation using the Completion/Cloze formulation (CF), ranking each answer string using language model probabilities. In terms of probability normalization, there is either no normalization (none) or normalization by the number of tokens in the answer (token) when ranking solely based on probability may heavily favor shorter answers (Brown et al., 2020). For MMLU, the in-loop evaluation also includes a setup where we increase the total number of instances by including a range of 0-shot to 5-shot setups together as we found this provides smoother trends as the training proceeds ("MMLU Var"). We also include the Multiple-choice formulation (MCF) version of MMLU, scoring prediction of answer labels like A/B/C/D, which generally starts to rise only later in training as models only gain the multiple-choice capability later (at around 1T tokens for **OLMOE-1B-7B** in Figure F3). We also evaluate perplexity on selected validation sets from Paloma (Magnusson et al., 2023; Reid et al., 2022; Gao et al., 2020; Soldaini et al., 2024; Liang et al., 2023; Merity et al., 2016). All code used for evaluation during pretraining is at `https://github.com/allenai/OLMo/tree/61ac104d616ec5435db225796e5c7532c9abd95a/olmo/eval`.

**After pretraining - OLMES**   We perform evaluations following the OLMES evaluation standard (Gu et al., 2024), with the suite of tasks in the original paper. OLMES (Open Language Model Evaluation Standard) is a standard for reproducible LM evaluations that is open, practical, and documented, providing recommendations guided by experiments and results from the literature (Biderman et al., 2024; Gao et al., 2021; Groeneveld et al., 2023). It is designed to support comparisons between smaller base models that require the Cloze formulation of multiple-choice questions against larger models that can utilize the Multiple-choice formulation. To make our evaluations reproducible, we follow OLMES in prompt formatting, choice of in-context examples, probability normalization, task formulation, as well as all other details. We summarize this setup in Table 2 and refer to Gu et al. (2024) for more details.

**After pretraining - DCLM**   For results on the DCLM tasks (Li et al., 2024a) in Table F2, we precisely follow their setup using the evaluation code released by the authors at `https://github.com/mlfoundations/dclm`. "Core" results are the `low variance` tasks in their evaluation code, while "Extended" corresponds to the `heavy` tasks.

**After adaptation**   After supervised finetuning and direct preference optimization, we evaluate models using a subset of the evaluations and the same overall setup used in Ivison et al. (2023) and Wang et al. (2023). We cover a wide range of model capabilities in our evaluation suite including coding (HumanEval Chen et al. (2021)), general and mathematical reasoning (Big Bench Hard Suzgun et al. (2022), GSM8k Cobbe et al. (2021)), world knowledge (MMLU), general instruction following (AlpacaEval 1.0 Li et al. (2023b), not the length-controlled variant (Dubois et al., 2024)), precise instruction following (IFEval Zhou et al. (2023b)) and safety (XSTest Röttger et al. (2024)). We refer to Wang et al. (2023) for more details on each benchmark.

# E   OPENNESS OF MODELS

We list the openness of various models summarized in Figure 1. We exclude Switch Transformers (Fedus et al., 2022), as it was published over three years ago and is very different from more recent MoE models (MLM objective, Encoder-decoder, etc.).

**Grok-86B-314B (xAI, 2024)**

- ✔ **Model:** Their model is licensed under the open-source Apache 2.0 license.
- ✗ **Data:** Unavailable.
- ✗ **Code:** Unavailable.
- ✗ **Logs:** Unavailable.

**Mixtral-39B-141B and Mixtral-13B-42B (Jiang et al., 2024)**

- ✔ **Model:** Their model is licensed under the open-source Apache 2.0 license.
- ✗ **Data:** Unavailable.
- ✗ **Code:** Unavailable.
- ✗ **Logs:** Unavailable.

**DBRX-36B-132B (Databricks, 2024)**

- ‼ **Model:** The model is licensed under a custom non-open-source license[8] with additional use-case restrictions.[9]
- ✗ **Data:** Unavailable.
- ✗ **Code:** They use closed-source custom adaptations of their public libraries LLM-foundry, composer, and megablocks.[10]
- ✗ **Logs:** Unavailable.

**Skywork-MoE-22B-146B (Wei et al., 2024)**

- ‼ **Model:** The model is licensed under a custom non-open-source license.[11]
- ✗ **Data:** Unavailable.

---

[8] `https://www.databricks.com/legal/open-model-license`
[9] `https://www.databricks.com/legal/acceptable-use-policy-open-model`
[10] `https://github.com/databricks/dbrx`
[11] `https://github.com/SkyworkAI/Skywork/blob/main/Skywork%20Community%20License.pdf`

- ✗ **Code:** Unavailable.
- ✗ **Logs:** Unavailable.

**DeepSeekV2-21B-236B (DeepSeek-AI et al., 2024b) and DeepSeekMoE-3B-14B (Dai et al., 2024)**

- ‼ **Model:** The models are licensed under custom non-open-source licenses.[12]
- ✗ **Data:** Unavailable.
- ✗ **Code:** Unavailable.
- ✗ **Logs:** Unavailable.

**Arctic-17B-480B (Snowflake, 2024a)**

- ✔ **Model:** The model is licensed under the open-source Apache 2.0 license.
- ‼ **Data:** They describe their mixture but do not release it.[13]
- ✗ **Code:** Unavailable.
- ✗ **Logs:** Unavailable.

**Qwen2-14B-57B (Team, 2024b)**

- ✔ **Model:** The model is licensed under the open-source Apache 2.0 license.
- ✗ **Data:** Unavailable.
- ✗ **Code:** Unavailable.
- ✗ **Logs:** Unavailable.

**Jamba-12B-52B (Lieber et al., 2024)**

- ✔ **Model:** The model is licensed under the open-source Apache 2.0 license.
- ✗ **Data:** Unavailable.
- ✗ **Code:** Unavailable.
- ✗ **Logs:** Unavailable.

**Qwen1.5-3B-14B (Team, 2024b)**

- ‼ **Model:** The model is licensed under a custom non-open-source license.[14]
- ✗ **Data:** Unavailable.
- ✗ **Code:** Unavailable.
- ✗ **Logs:** Unavailable.

**JetMoE-2B-9B (Shen et al., 2024)**

- ✔ **Model:** The model is licensed under the open-source Apache 2.0 license.
- ‼ **Data:** They describe their mixture but do not release it.
- ‼ **Code:** They make their fork of megablocks publicly available,[15] however, their Megatron-LM training code is not available.[16]
- ✗ **Logs:** Unavailable.

---

[12] https://github.com/deepseek-ai/DeepSeek-MoE/blob/main/LICENSE-MODEL and https://github.com/deepseek-ai/DeepSeek-V2/blob/main/LICENSE-MODEL

[13] https://medium.com/snowflake/snowflake-arctic-cookbook-series-arctics-approach-to-data-b81

[14] https://hf.co/Qwen/Qwen1.5-MoE-A2.7B/blob/main/LICENSE

[15] https://github.com/yikangshen/megablocks

[16] https://hf.co/jetmoe/jetmoe-8b/discussions/5#661ee52c03251697a0b155cc

**OpenMoE-2B-9B (Xue et al., 2024)**

- ✔ **Model:** The model is licensed under the open-source Apache 2.0 license.
- ✔ **Data:** They make scripts for recreating their data available.
- ✔ **Code:** They make their code available.[17]
- ✘ **Logs:** Unavailable.

**OLMOE-1B-7B**

- ✔ **Model:** The model is licensed under the open-source Apache 2.0 license.
- ✔ **Data:** The data is licensed under the open-source ODC-By 1.0 license.
- ✔ **Code:** The code is licensed under the open-source Apache 2.0 license.
- ✔ **Logs:** Logs are available with the same open-source license as the code (Apache 2.0).

# F    ADDITIONAL EVALUATION

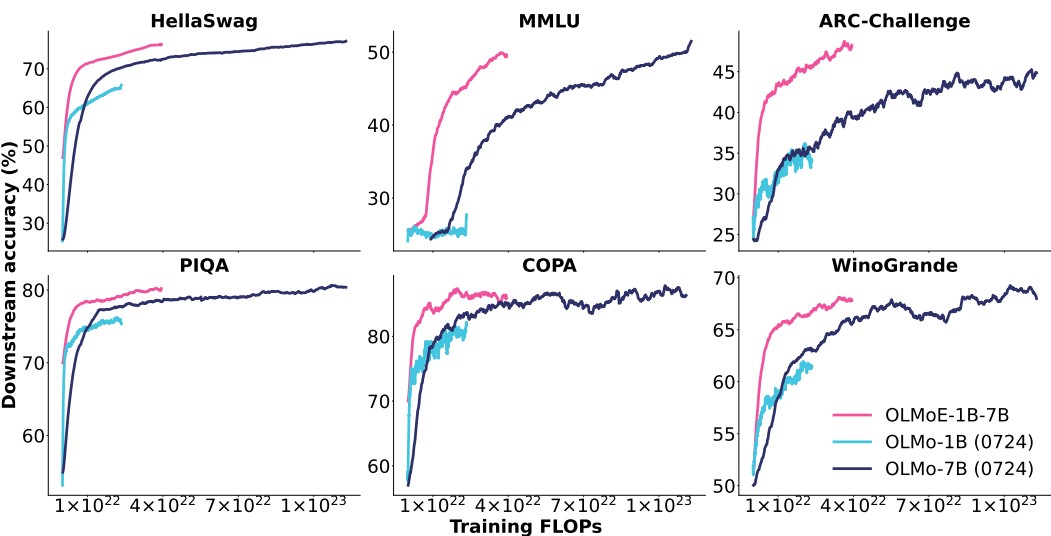

Figure F1: **Evaluation of OLMOE-1B-7B and the current best OLMo models during pretraining.** OLMOE-1B-7B differs from the OLMo models in its MoE architecture, several training hyperparameters, and its training dataset, see §2. A version of this plot with tokens as the x-axis and markers where annealing starts is in Appendix F. More results, logs, and configurations: https://wandb.ai/ai2-llm/olmoe/reports/Plot-OLMoE-1B-7B-vs-OLMo-7B-vs-OLMo-1B--Vmlldzo4OTcyMjEz

**During pretraining** In Figure F1 we benchmark the performance of **OLMOE-1B-7B** during pretraining with the current best OLMo models (Groeneveld et al., 2024) on commonly used downstream tasks. We find that across all tasks **OLMOE-1B-7B** reaches better performance with less compute (FLOPs) than the dense OLMo models. **OLMOE-1B-7B** matches or outperforms OLMo-7B at the end of training despite **OLMOE-1B-7B** having used less than half as many FLOPs for training and using only 1B active parameters. This is likely a result of the dataset and modeling changes we make to the OLMo setup including MoE-related changes, stability, and performance improvements, outlined in Appendix C. Appendix F contains training and validation loss plots showing very smooth loss curves without major loss spikes during the 5T tokens of our pretraining.

---

[17]https://github.com/XueFuzhao/OpenMoE/tree/main?tab=readme-ov-file#training-with-tpugpu

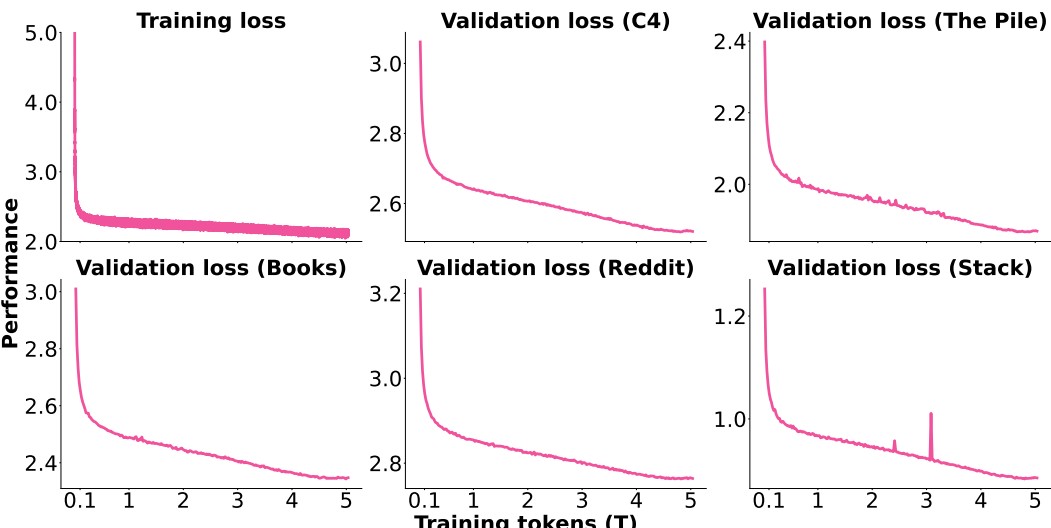

Figure F2: **Losses of OLMOE-1B-7B during training.** The Books, Reddit, and Stack (Kocetkov et al., 2022) datasets are from Dolma 1.7 (Soldaini et al., 2024) via Paloma (Magnusson et al., 2023). More results, logs, and configurations: https://wandb.ai/ai2-llm/olmoe/reports/Plot-OLMoE-1B-7B--Vmlldzo4OTcyMjU3

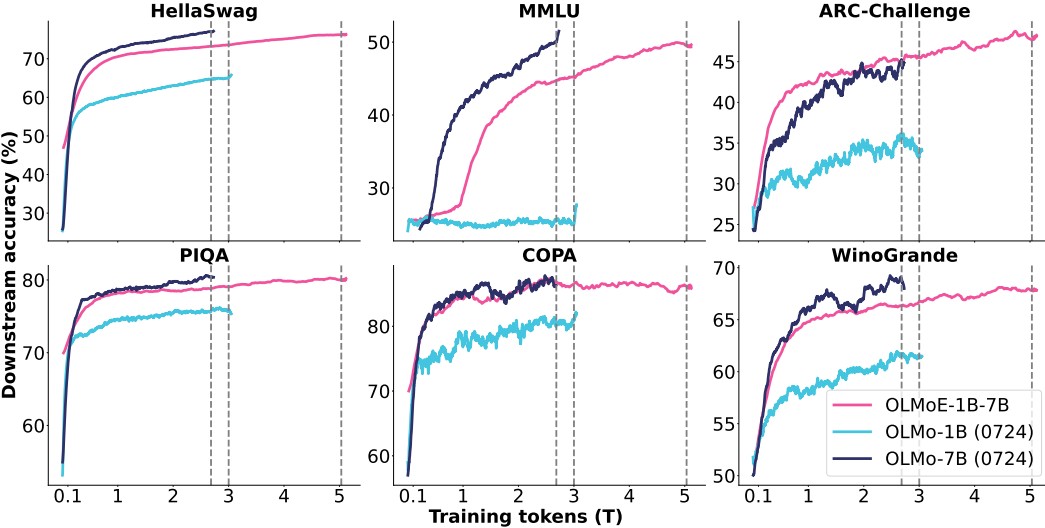

Figure F3: **Evaluation of OLMOE-1B-7B and the current best OLMo models during pretraining.** Grey vertical lines correspond to where the respective run enters annealing with the 1st line being for OLMo-7B, the 2nd for OLMo-1B, and the third for **OLMOE-1B-7B**. Figure F1 is a version of this plot with training FLOPs as the x-axis. More results, logs, and configurations: https://wandb.ai/ai2-llm/olmoe/reports/Plot-OLMoE-1B-7B-vs-OLMo-7B-vs-OLMo-1B--Vmlldzo4OTcyMjEz

Table F1: More results on OLMES. $^{\dagger}$ indicates use of the MCF score, see Appendix D. See Table 2 for details on naming and a summary of these results.

| Model | ARC_C | ARC_E | BoolQ | CSQA | HSwag | MMLU | OBQA | PIQA | SIQA | WinoG | Avg |
|---|---|---|---|---|---|---|---|---|---|---|---|
| *LMs with ~7-9B active parameters* | | | | | | | | | | | |
| Mistral-7B | 78.6$^{\dagger}$ | 90.8$^{\dagger}$ | 89.3 | 72.4$^{\dagger}$ | 83.0 | 64.0$^{\dagger}$ | 80.6$^{\dagger}$ | 82.8 | 71.3$^{\dagger}$ | 77.9 | 79.1 |
| OLMo-7B (0724) | 68.0$^{\dagger}$ | 85.7$^{\dagger}$ | 85.3 | 85.4$^{\dagger}$ | 80.5 | 54.9$^{\dagger}$ | 67.6$^{\dagger}$ | 79.3 | 76.1$^{\dagger}$ | 73.2 | 75.6 |
| DCLM-7B | 79.8$^{\dagger}$ | 92.3$^{\dagger}$ | 87.0 | 77.0 | 82.3 | 64.4$^{\dagger}$ | 79.6$^{\dagger}$ | 80.1 | 71.2$^{\dagger}$ | 77.3 | 79.1 |
| Llama2-7B | 54.2 | 84.0 | 86.1 | 74.2 | 78.9 | 46.2$^{\dagger}$ | 57.8 | 77.5 | 59.6 | 71.7 | 69.0 |
| Llama3.1-8B | 79.5$^{\dagger}$ | 91.7$^{\dagger}$ | 88.5 | 74.3$^{\dagger}$ | 81.6 | 66.9$^{\dagger}$ | 78.6$^{\dagger}$ | 81.1 | 71.4$^{\dagger}$ | 76.6 | 79.0 |
| Gemma2-9B | 89.5$^{\dagger}$ | 95.5$^{\dagger}$ | 89.4 | 78.8$^{\dagger}$ | 87.3$^{\dagger}$ | 70.6$^{\dagger}$ | 88.4$^{\dagger}$ | 86.1$^{\dagger}$ | 76.0$^{\dagger}$ | 78.8 | 84.0 |
| *LMs with ~2-3B active parameters* | | | | | | | | | | | |
| StableLM-2B | 50.6$^{\dagger}$ | 75.3 | 82.3 | 70.4$^{\dagger}$ | 70.3 | 40.4$^{\dagger}$ | 56.6$^{\dagger}$ | 75.6 | 64.3$^{\dagger}$ | 65.8 | 65.1 |
| Gemma2-3B | 67.5$^{\dagger}$ | 84.3$^{\dagger}$ | 83.6 | 66.4$^{\dagger}$ | 74.6 | 53.3$^{\dagger}$ | 68.8$^{\dagger}$ | 78.5 | 64.7$^{\dagger}$ | 71.8 | 71.4 |
| JetMoE-2B-9B | 61.4$^{\dagger}$ | 81.9$^{\dagger}$ | 85.7 | 75.3$^{\dagger}$ | 81.7 | 49.1$^{\dagger}$ | 68.0$^{\dagger}$ | 80.3 | 71.3$^{\dagger}$ | 70.7 | 72.5 |
| OpenMoE-3B-9B | 29.3 | 50.6 | 63.2 | 21.5 | 44.4 | 27.4 | 34.6 | 63.3 | 42.9 | 51.9$^{\dagger}$ | 42.9 |
| DeepSeek-3B-16B | 53.4 | 82.7 | 81.9 | 72.7 | 80.4 | 45.5$^{\dagger}$ | 58.4 | 80.1 | 59.9 | 73.2 | 68.8 |
| Llama3.2-3B | 69.6$^{\dagger}$ | 85.1$^{\dagger}$ | 78.3 | 69.0 | 77.0 | 57.8$^{\dagger}$ | 67.2$^{\dagger}$ | 77.4 | 64.9$^{\dagger}$ | 69.9 | 71.6 |
| Qwen1.5-3B-14B | 77.4$^{\dagger}$ | 91.6$^{\dagger}$ | 85.0 | 81.4$^{\dagger}$ | 80.0 | 62.4$^{\dagger}$ | 80.6$^{\dagger}$ | 81.0 | 74.1$^{\dagger}$ | 72.3 | 78.6 |
| *LMs with ~1B active parameters* | | | | | | | | | | | |
| OLMo-1B (0724) | 36.4 | 53.5 | 66.8 | 42.4 | 67.5 | 32.1 | 44.2 | 74.0 | 45.2 | 62.9 | 52.5 |
| TinyLlama-1B | 38.1 | 69.5 | 63.6 | 61.1 | 60.8 | 33.6 | 45.0 | 71.7 | 50.4 | 60.1 | 55.4 |
| Pythia-1B | 31.4 | 63.4 | 56.8$^{\dagger}$ | 50.9 | 48.0 | 31.1 | 40.4 | 68.9 | 46.4 | 52.7 | 49.0 |
| Llama3.2-1B | 43.5 | 71.6 | 69.4 | 59.6 | 67.3 | 38.2 | 42.0 | 73.7 | 52.0 | 62.5 | 58.0 |
| Zamba2-1B | 55.0$^{\dagger}$ | 85.4 | 76.1 | 70.1 | 73.4 | 44.73$^{\dagger}$ | 59.8$^{\dagger}$ | 76.6 | 58.4 | 67.2 | 66.7 |
| DCLM-1B | 57.6$^{\dagger}$ | 79.5 | 80.9 | 71.3 | 75.1 | 48.5$^{\dagger}$ | 60.0$^{\dagger}$ | 76.6 | 60.5$^{\dagger}$ | 68.1 | 67.8 |
| **OLMoE-1B-7B** | **62.1$^{\dagger}$** | **84.2** | **79.2** | **72.9** | **80.0** | **54.1$^{\dagger}$** | **65.4$^{\dagger}$** | **79.8** | **63.0$^{\dagger}$** | **70.2** | **71.1** |

Table F2: **DCLM evaluation metrics on the Core and Extended task subsets (Li et al., 2024a).**
*=Core tasks. "annealed" is the final pretraining checkpoint we use for **OLMoE-1B-7B** and was annealed from the checkpoint at step 1,200,000. We left the non-annealing pretraining run train a little longer resulting in the 1,220,000 checkpoint.

| OLMoE-1B-7B checkpoint (→) | step 1,200,000 | step 1,220,000 | annealed | OLMo-1B | OLMo-7B |
|---|---|---|---|---|---|
| AGI Eval LSAT-AR* | 24.3 | 26.5 | 28.7 | 28.3 | 28.3 |
| AGI Eval LSAT-LR | 40.2 | 38.6 | 37.3 | 30.2 | 42.9 |
| AGI Eval LSAT-RC | 47.4 | 43.7 | 46.6 | 23.5 | 61.6 |
| AGI Eval SAT-En | 55.3 | 54.9 | 52.9 | 28.2 | 73.8 |
| AGI Eval SAT-Math CoT | 5.5 | 4.1 | 6.4 | 1.8 | 6.8 |
| AQuA CoT | 2.4 | 2.9 | 2.0 | 2.9 | 6.1 |
| ARC Challenge* | 53.3 | 53.4 | 53.8 | 34.6 | 48.1 |
| ARC Easy* | 77.1 | 78.5 | 77.7 | 64.4 | 75.9 |
| BBQ | 49.8 | 48.3 | 50.6 | 45.8 | 67.2 |
| BigBench CS Algorithms* | 47.1 | 50.2 | 47.2 | 47.5 | 53.6 |
| BigBench Conceptual Combinations | 51.5 | 50.5 | 56.3 | 31.1 | 68.0 |
| BigBench Conlang Translation | 3.7 | 6.1 | 7.3 | 4.3 | 7.3 |
| BigBench Dyck Languages* | 19.3 | 15.9 | 21.5 | 26.6 | 22.2 |
| BigBench Elementary Math QA | 26.2 | 27.0 | 26.9 | 26.2 | 30.4 |
| BigBench Language Identification* | 31.9 | 34.0 | 31.0 | 27.0 | 39.1 |
| BigBench Logical Deduction | 26.6 | 25.3 | 24.6 | 23.6 | 27.3 |
| BigBench Misconceptions | 59.8 | 55.3 | 62.6 | 55.7 | 58.0 |
| BigBench Novel Concepts | 62.5 | 62.5 | 65.6 | 43.8 | 53.1 |
| BigBench Operators* | 36.2 | 34.3 | 33.8 | 23.8 | 45.2 |
| BigBench QA Wikidata* | 68.2 | 68.8 | 69.2 | 67.0 | 69.9 |
| BigBench Repeat Copy Logic* | 15.6 | 15.6 | 18.8 | 3.1 | 9.4 |
| BigBench Strange Stories | 66.7 | 68.4 | 69.5 | 53.4 | 66.1 |
| BigBench Strategy QA | 56.2 | 58.1 | 57.0 | 51.5 | 68.6 |
| BigBench Understanding Fables | 47.1 | 44.4 | 47.6 | 28.0 | 61.4 |
| BoolQ* | 73.3 | 72.8 | 73.2 | 63.7 | 83.9 |
| COPA* | 81.0 | 80.0 | 78.0 | 75.0 | 77.0 |
| CoQA* | 43.7 | 44.4 | 43.7 | 3.4 | 45.4 |
| CommonsenseQA* | 67.2 | 67.0 | 69.3 | 19.6 | 86.0 |
| Enterprise PII Classification | 52.3 | 53.7 | 52.2 | 57.3 | 50.6 |
| GPQA Diamond | 22.2 | 21.2 | 19.7 | 19.7 | 20.2 |
| GPQA Main | 24.8 | 22.3 | 22.5 | 20.3 | 23.0 |
| GSM8K CoT | 6.4 | 7.4 | 7.4 | 4.9 | 30.6 |
| HellaSwag 0-shot* | 76.0 | 76.0 | 77.0 | 65.8 | 76.7 |
| HellaSwag 10-shot* | 77.6 | 77.5 | 78.6 | 66.3 | 78.9 |
| Jeopardy* | 48.8 | 48.7 | 50.3 | 22.6 | 46.5 |
| LAMBADA* | 72.7 | 72.2 | 73.3 | 61.1 | 71.8 |
| LogiQA | 34.9 | 34.3 | 34.6 | 28.7 | 31.0 |
| MMLU Few-shot | 52.2 | 51.9 | 53.3 | 28.4 | 55.1 |
| MMLU Zero-shot | 41.6 | 42.7 | 43.3 | 26.2 | 50.0 |
| Math QA | 26.4 | 27.1 | 27.5 | 24.1 | 29.8 |
| OpenBookQA* | 41.4 | 44.0 | 44.8 | 36.6 | 43.4 |
| PIQA* | 81.3 | 81.2 | 82.0 | 76.4 | 81.7 |
| PubMedQA | 56.1 | 46.6 | 57.9 | 0.2 | 57.9 |
| SQuAD* | 52.9 | 52.4 | 52.4 | 0.0 | 65.5 |
| SVAMP CoT | 30.0 | 28.0 | 33.0 | 14.3 | 44.7 |
| Simple Arithmetic, no spaces | 17.6 | 18.1 | 20.1 | 1.2 | 15.3 |
| Simple Arithmetic, with spaces | 19.5 | 20.6 | 22.1 | 1.8 | 16.0 |
| Social IQA | 71.5 | 70.7 | 69.3 | 69.5 | 84.4 |
| Trivia QA | 54.2 | 53.0 | 55.9 | 25.1 | 51.8 |
| Winogender Female | 50.0 | 46.7 | 50.0 | 41.7 | 58.3 |
| Winogender Male | 55.0 | 58.3 | 60.0 | 63.3 | 58.3 |
| Winograd* | 82.8 | 83.2 | 84.6 | 79.9 | 83.2 |
| Winogrande* | 68.0 | 68.5 | 69.0 | 61.8 | 67.6 |
| Core | 46.3 | 46.5 | 47.2 | 30.2 | 49.8 |
| Extended | 31.3 | 30.9 | 32.5 | 16.9 | 37.0 |

# G  OTHER EXPERIMENTS

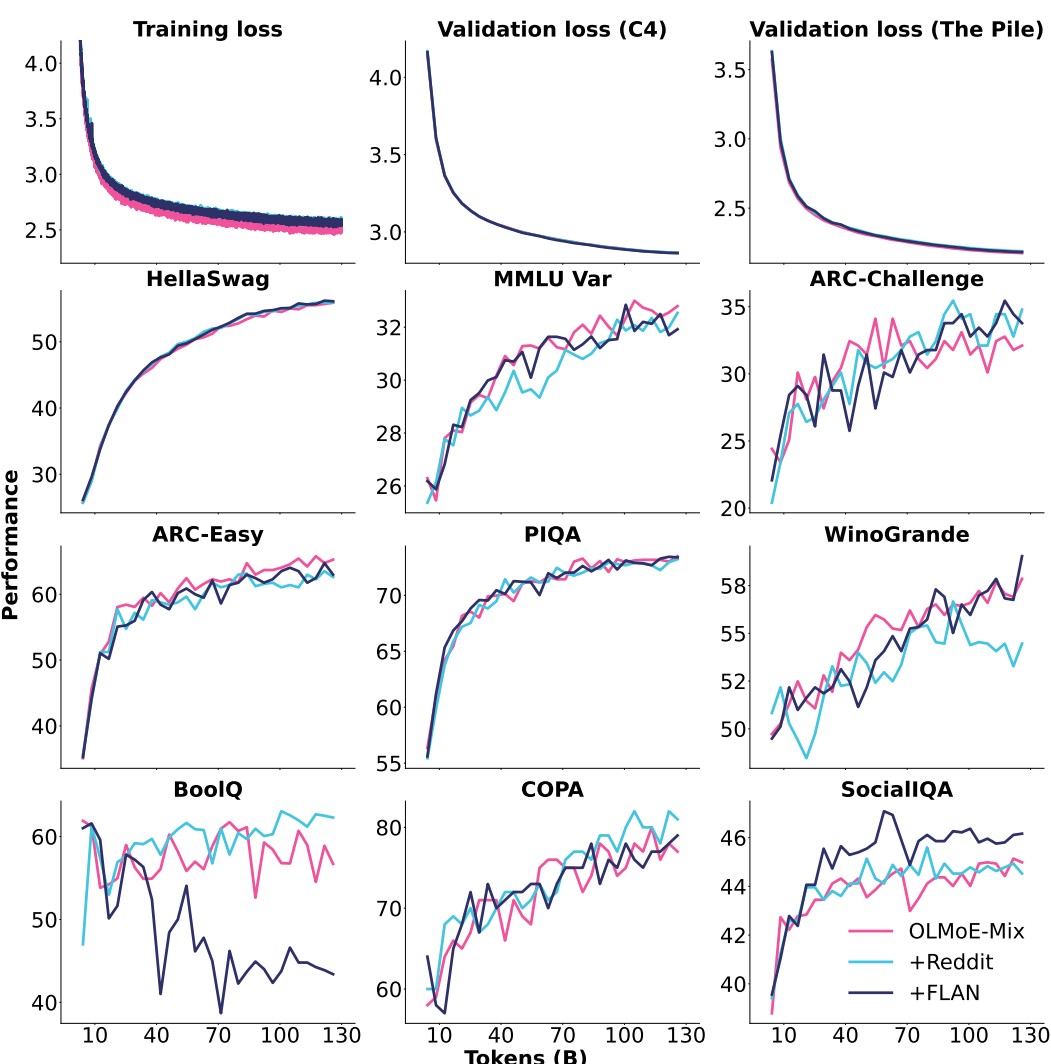

Figure G1: **Adding Reddit or FLAN to OLMOE-MIX.** More results, logs, and configurations: https://wandb.ai/ai2-llm/olmoe/reports/Plot-Adding-Reddit-FLAN--Vmlldzo4OTg1NTg4

**Adding Reddit or FLAN to OLMOE-MIX**  In Figure G1 we benchmark adding the Reddit or FLAN (Wei et al., 2022) subsets of Dolma 1.7 (Soldaini et al., 2024) to our pretraining data mix (§2). Overall, we do not find either one to lead to consistent gains, thus we do not use them in our final data mix.

**Load balancing precision**  Fedus et al. (2022) selectively perform operations related to routing in full precision (FP32) to improve stability. In Figure G2, we test whether computing the load balancing loss in full precision improves stability, but do not find it to reduce spikes. Thus, we stick with bfloat16 (BF16).

**Noise upcycling**  For the creation of Qwen2-MoE (Yang et al., 2024a; Team, 2024b; Bai et al., 2023a), the authors add 50% of gaussian noise to feedforward networks before continuing training in an upcycled setup (Komatsuzaki et al., 2023). Komatsuzaki et al. (2023) also report that they experimented with adding noise but did not find it beneficial. In Figure G3, we experiment with

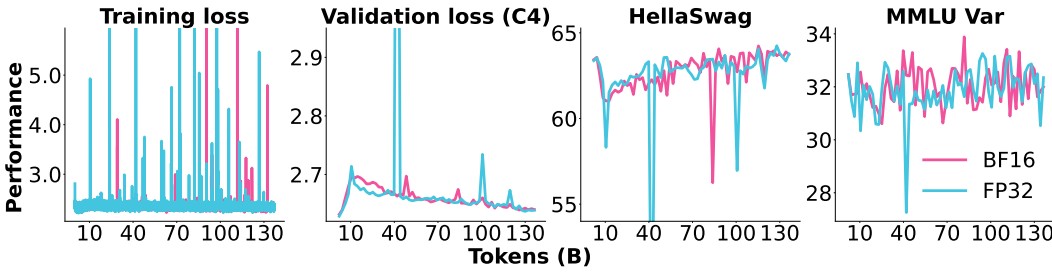

Figure G2: **Load balancing precision.** More results, logs, and configurations: `https://wandb.ai/ai2-llm/olmoe/reports/Plot-FP32-LBL--Vmlldzo4NDMxNDA4`

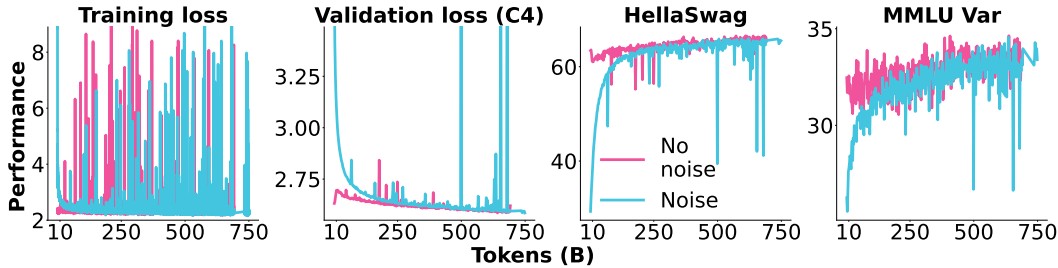

Figure G3: **Adding noise to the upcycled checkpoint.** More results, logs, and configurations: `https://wandb.ai/ai2-llm/olmoe/reports/Plot-Noise-upcycle---Vmlldzo4NDA3MzI2`

regular upcycling versus adding noise by randomly replacing 50% of each MLP with numbers drawn from a normal distribution with a standard deviation of 0.02 following. We find that after 700 billion tokens, the no noise variant still performs slightly better but both appear to converge to the same performance. If training further, it is possible that the noise variant eventually outperforms the no noise variant, but at that point, it may make more sense to just train the MoE from scratch (§B.1.2).

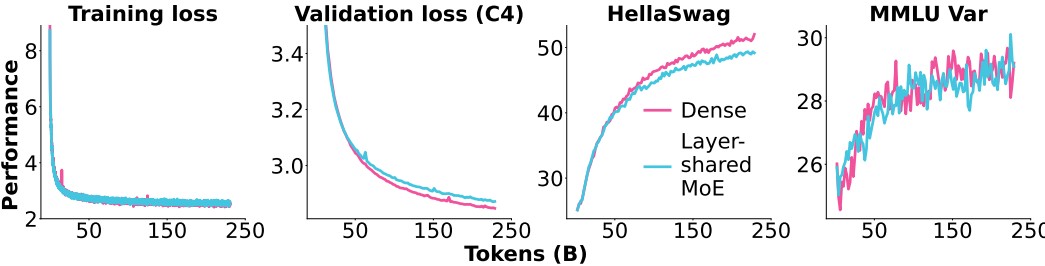

Figure G4: **Sharing the same MoE across layers versus a regular dense LM.** The number of experts in the MoE is equivalent to its number of layers. Thus, because the MoE is shared across layers, it has the same number of total and active parameters as the dense model. More results, logs, and configurations: `https://wandb.ai/ai2-llm/olmoe/reports/Plot-Shared-vs-Dense--Vmlldzo4NDI0MTc5`

**Shared Layer**   Some work has investigated Mixture-of-Experts with weights shared across layers in the context of Universal Transformers (Tan et al., 2023; Csordás et al., 2024; Dehghani et al., 2019). We test whether layer-shared Mixture-of-Experts can beat non-shared dense models in Figure G4. The layer-shared MoE uses a load balancing loss that is applied at the model level rather than at the layer level. This gives the model more flexibility by allowing it to completely deactivate certain experts for some layers and even emulate a dense model by always activating one separate expert for

each layer. This makes it a generalization of the dense model which motivated our hypothesis that it may perform better than the dense model. However, in practice, we find that both perform similarly with the regular dense models even maintaining a small advantage on validation loss and HellaSwag. One possible advantage of layer-shared MoEs is that they can allow for better load balancing at inference. If prompts come in continuously, then newly incoming prompts can be batched with previous prompts that have already passed through several layers and sent through the MoE module together, as the MoE module is the same regardless of whether it is the first or last layer. Sharing also reduces throughput by around 20% during training, which further motivates our decision not to use it for **OLMoE-1B-7B**.

**KTO experiments** In Table G1 we experiment with the number of steps (5,000 vs. 10,000) and the optimizer (Adam (Kingma & Ba, 2017) vs. RMS) used for KTO (Ethayarajh et al., 2024). Based on these experiments we use the RMS optimizer and the checkpoint at 5,000 steps in §B.3.

Table G1: **KTO adaptation experiments.** 5,000 and 10,000 steps correspond to 1.3 and 2.6 epochs on our adaptation dataset (§2), respectively.

| Task ($\rightarrow$)
Setup ($\rightarrow$)
Metric ($\rightarrow$) | MMLU
0-shot
EM | GSM8k
8-shot CoT
EM | BBH
0-shot
EM | Human-
Eval
0-shot
Pass@10 | Alpaca-
Eval 1.0
0-shot
%win | XSTest
0-shot
F1 | IFEval
0-shot
Loose Acc | Avg
0-shot |
|---|---|---|---|---|---|---|---|---|
| KTO, 5,000 steps, RMS | **51.2** | **45.5** | 34.1 | **57.1** | **81.6** | **86.6** | **47.5** | **57.7** |
| KTO, 10,000 steps, RMS | 51.0 | 41.0 | 34.7 | 53.8 | 81.0 | 62.3 | **47.5** | 54.2 |
| KTO, 5,000 steps, Adam | **51.2** | 42.0 | **35.3** | 55.6 | 81.0 | 84.5 | 46.6 | 56.0 |
| KTO, 10,000 steps, Adam | 51.0 | 43.0 | 34.1 | 54.9 | 79.7 | 62.7 | **47.5** | 53.3 |

## H   ANALYSIS

### H.1   DETAILS OF ANALYSIS IN §5

**Router saturation** We define router saturation as the proportion of expert activations at some intermediary checkpoint at time $t$ that matches the expert IDs activated at some final checkpoint over the same dataset:

$$\text{Router Saturation}(t) = \frac{1}{N} \sum_{i=1}^{N} \frac{|\mathcal{E}_i^{(t)} \cap \mathcal{E}_i^{(T)}|}{k}, \qquad (5)$$

where:

- $N$: The total number of tokens in the dataset.

- $k$: The number of top-$k$ experts activated per input token. While we train with $k = 8$ (§2), we also analyze $k = 1$ by only looking at the expert with the highest routing probability.

- $\mathcal{E}_i^{(t)}$: The set of $k$ experts activated for the $i$th token at the $t$th checkpoint.

- $\mathcal{E}_i^{(T)}$: The set of $k$ experts activated for the $i$th token at the final checkpoint $T$.

- $|\mathcal{E}_i^{(t)} \cap \mathcal{E}_i^{(T)}|$: The number of common experts activated for the $i$th token between the $t$th and final checkpoints.

Router saturation thus corresponds to whether the router weights are still learning which expert will process certain data. A value of 100% indicates that the router at the intermediate checkpoint will route to the same experts as the final checkpoint router. However, even at 100% saturation the router weight can still change and adapt the exact router probability for each expert. These probabilities are used to scale the output of the respective expert in the model. For **OLMoE-1B-7B** with its 64 experts, random routing equals a saturation of $1/64 = 1.6\%$ for $k = 1$ and $8/64 = 12.5\%$ for $k = 8$.

**Expert co-activation**   We define expert co-activation as the proportion of times two specific experts, $E_i$ and $E_j$, are simultaneously activated out of the total number of activations of one of those experts:

$$\text{Expert co-activation}(E_i, E_j) = \frac{N_{E_i, E_j}}{N_{E_i}}, \tag{6}$$

where:

- $E_i$: The first expert.
- $E_j$: The second expert.
- $N_{E_i, E_j}$: The number of times experts $E_i$ and $E_j$ are activated together.
- $N_{E_i}$: The total number of times expert $E_i$ is activated.

A co-activation of 100% indicates that if $E_i$ is activated, $E_j$ is also always activated. A value of 0% indicates that the experts never co-occur. If multiple expert pairs have high co-activation, it may suggest that these experts could be merged, benefiting less from keeping them separate. In a distributed setup, we could place highly co-activated experts on the same device to reduce communication costs during model inference.

**Domain specialization**   We define domain specialization as the proportion of tokens from a particular domain $D$ that get routed to a particular expert $E_i$:

$$\text{Domain specialization}(E_i, D) = \frac{N_{E_i, D}^{(k)}}{N_D}, \tag{7}$$

where:

- $E_i$: The $i$th expert in the model.
- $D$: The domain from which the data originates.
- $k$: The number of experts considered (e.g., $k = 8$ means considering the top 8 experts with the highest routing probabilities).
- $N_{E_i, D}^{(k)}$: The number of tokens from domain $D$ for which $E_i$ is among the top-$k$ selected experts.
- $N_D$: The total number of tokens from domain $D$ processed by the MoE.

Domain specialization thus refers to the specialization of expert $E_i$ to domain $D$. A value of 100% indicates that all data from that domain is routed to $E_i$, whereas 0% indicates the expert is never used for that domain and can be removed from the model without affecting performance in that domain.

**Vocabulary specialization**   We define vocabulary specialization as the proportion of tokens with a token ID $x$ (also called vocabulary element) that are routed to one particular expert $E_i$ out of all experts in that layer:

$$\text{Vocabulary specialization}(E_i, x) = \frac{N_{x, E_i}^{(k)}}{N_x}, \tag{8}$$

where:

- $E_i$: The $i$th expert in the model.
- $x$: The token ID being analyzed.
- $k$: The number of experts considered (e.g., $k = 8$ means considering the top 8 experts with the highest routing probabilities).
- $N_{x, E_i}$: The number of times input data is routed to $E_i$ for $x$.
- $N_x$: The total number of times input data is routed across all experts for $x$.

Vocabulary specialization thus refers to how specialized a particular expert is on some vocabulary item. We distinguish input and output variants of this specialization, where $x$ is either the input token ID or the next output token ID (either the ground-truth next token ID or the token ID predicted by the model). A value of 100% indicates that for all occurrences of that vocabulary element, input data is routed to $E_i$, whereas 0% indicates an expert that is fully irrelevant for that vocabulary element and can be effectively removed from the model without affecting performance whenever the token ID appears.

## H.2 ADDITIONAL ANALYSIS

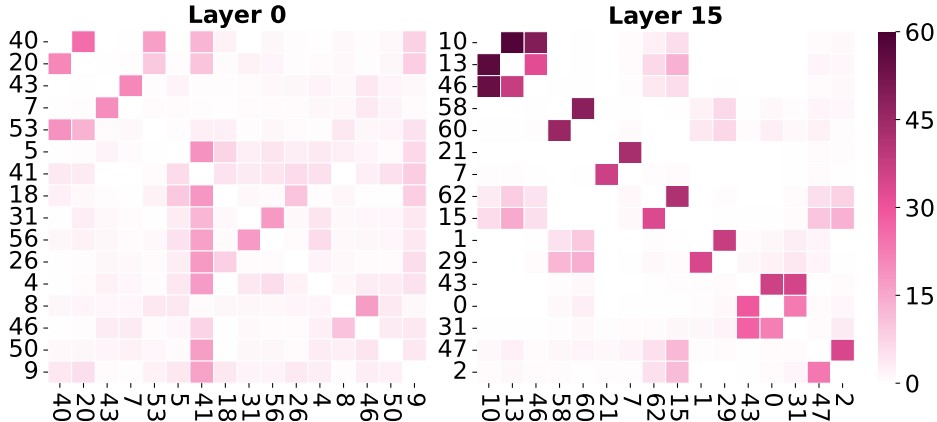

Figure H1: **Co-activation among experts of OLMoE-1B-7B on a random 0.5% of the C4 validation data.** We display the 32 experts with the highest maximum co-activation score via their expert IDs on the x- and y-axis. See Figure 6 for layer 7.

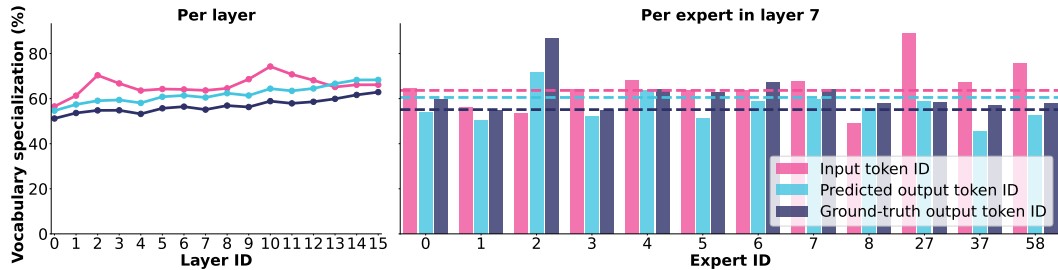

Figure H2: **Vocabulary specialization of OLMoE-1B-7B across layers and experts.** To compute vocabulary specialization per layer (left) we average the specialization of each expert in that layer. Dashed lines (right) correspond to the average of layer 7 as depicted left. We display the first 32 experts out of 64. This plot is when $k = 1$ in Equation 8.

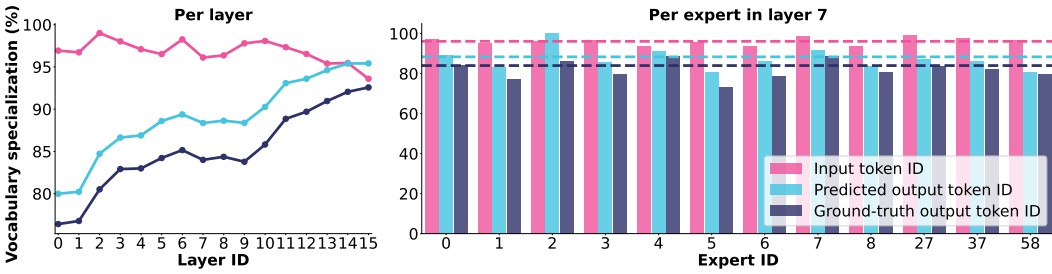

Figure H3: **Vocabulary specialization for OLMoE-1B-7B when considering all 8 activated experts.** Equivalent to $k = 8$ in Equation 8.

Table H1: **Vocabulary specialization in the 7th layer of OLMoE-1B-7B.** We use $k = 1$ (Equation 8) and a random 0.5% of the C4 validation data excluding token IDs with <10 appearances. See Table 4 for more.

| Expert ID | Input token IDs | Predicted output token IDs |
|---|---|---|
| 58 | (" (100%) (" (100%) ' (94%) ' (92%) " (92%) ( (92%) " (90%) ' (89%) " (88%) $ (87%) [ (87%) £ (86%) | such (100%) 486 (100%) see (95%) which (91%) driving (91%) UK (90%) who (88%) including (88%) normal (88%) |
| 7 | Him (100%) inde (100%) Jesus (98%) God (90%) pray (81%) Holy (80%) Quran (80%) God (77%) Lord (76%) glory (75%) Spirit (66%) Christ (65%) | rella (100%) Him (94%) sin (90%) prince (80%) glory (72%) Jesus (69%) Lord (68%) Christ (65%) Spirit (55%) Holy (53%) God (50%) Prayer (50%) |
| 37 | Sunday (100%) Tuesday (100%) Thursday (100%) Olympic (100%) Christmas (100%) rugby (100%) Championship (100%) weekends (100%) | days (91%) anniversary (90%) month (88%) week (84%) mpi (83%) semester (81%) mand (80%) Olympics (78%) cent (76%) season (76%) perm (75%) |
| 0 | ESM (100%) icillin (100%) agra (98%) aust (96%) asa (93%) pills (92%) mg (90%) uk (82%) login (82%) doc (81%) generic (81%) cd (81%) Essay (81%) password (81%) Content (80%) | *, (100%) sil (96%) pills (91%) vi (96%) xen (87%) pharmacy (87%) gener (85%) aust (82%) mg (75%) Content (75%) uk (73%) THAT (73%) dispens (68%) icillin (68%) generic (66%) |
| 3 | grandmother (92%) brother (91%) Daisy (83%) daughter (78%) mum (75%) father (72%) wife (70%) husband (70%) lady (63%) dad (62%) boy (61%) | hood (36%) mother (35%) inde (31%) boy (29%) girl (28%) married (27%) tri (21%) Gab (20%) died (18%) taught (14%) lived (13%) knew (10%) |

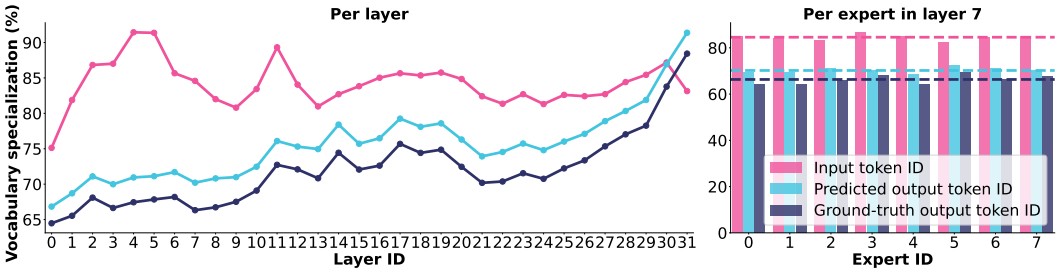

Figure H4: **Vocabulary specialization for Mixtral-8x7B when considering all 2 activated experts.** Equivalent to $k = 2$ in Equation 8.

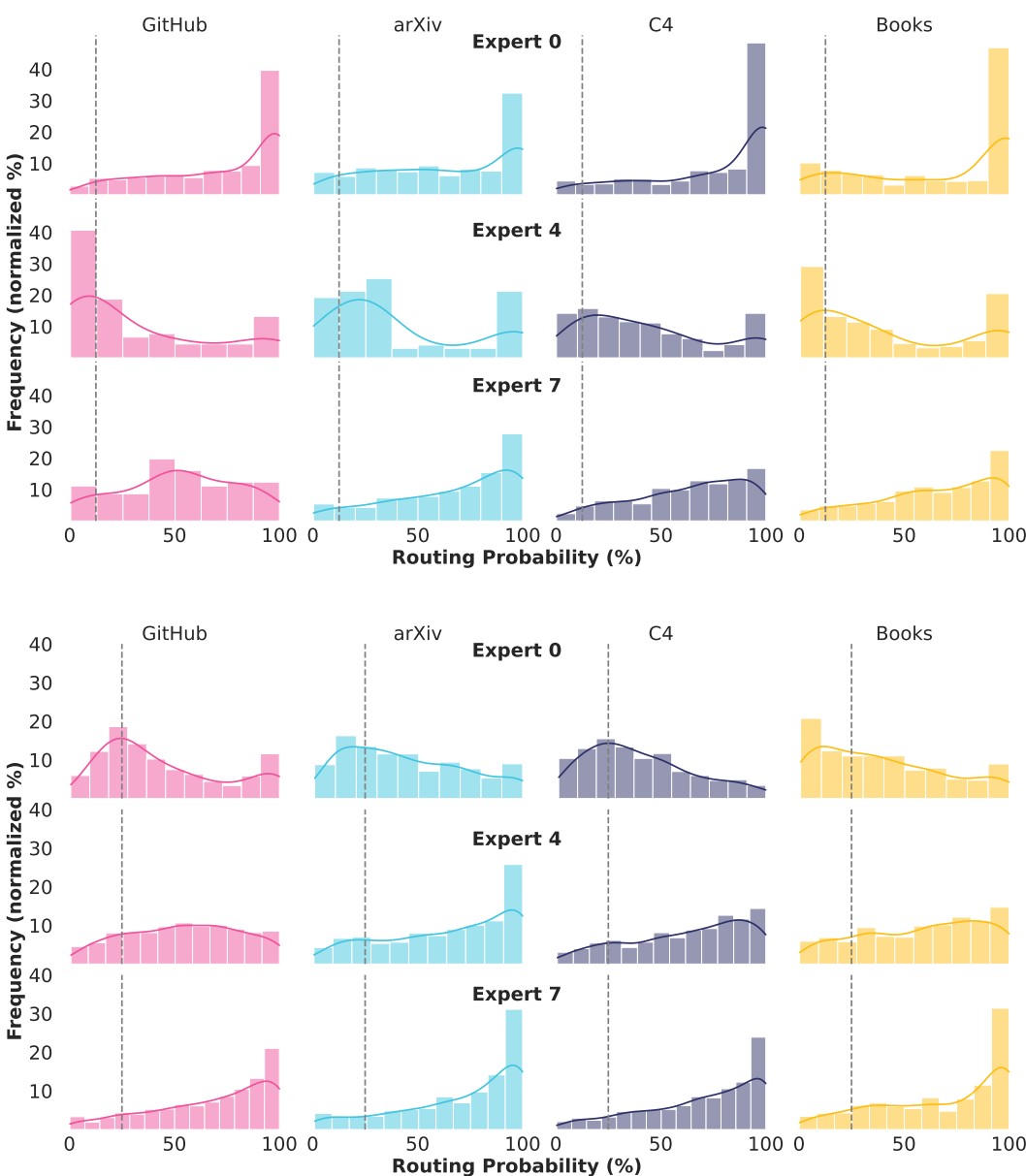

Figure H5: **Vocabulary specialization across domains of OLMoE-1B-7B (top) and Mixtral-8x7B (bottom).** We visualize how often token IDs get routed to specific experts. We only include IDs that appear at least 8 times in the various corpora. Vertical gray lines correspond to uniform routing (8/64=12.5% for **OLMoE-1B-7B** as it has 64 experts, 8 of which are activated; 2/8=25% for Mixtral as it has 8 experts, 2 of which are activated). For example, among all token IDs in GitHub that get routed to Expert 0 at least 8 times for **OLMoE-1B-7B**, ~40% of them get routed to Expert 0 with a probability of ~100% (upper left) indicating that Expert 0 is specialized on those token IDs. For **OLMoE-1B-7B** there is much frequency at the routing probability extremes (0% or 100%) indicating that these experts exclusively focus on certain token IDs, especially for *specific domains* (§5.3) like GitHub and arXiv.

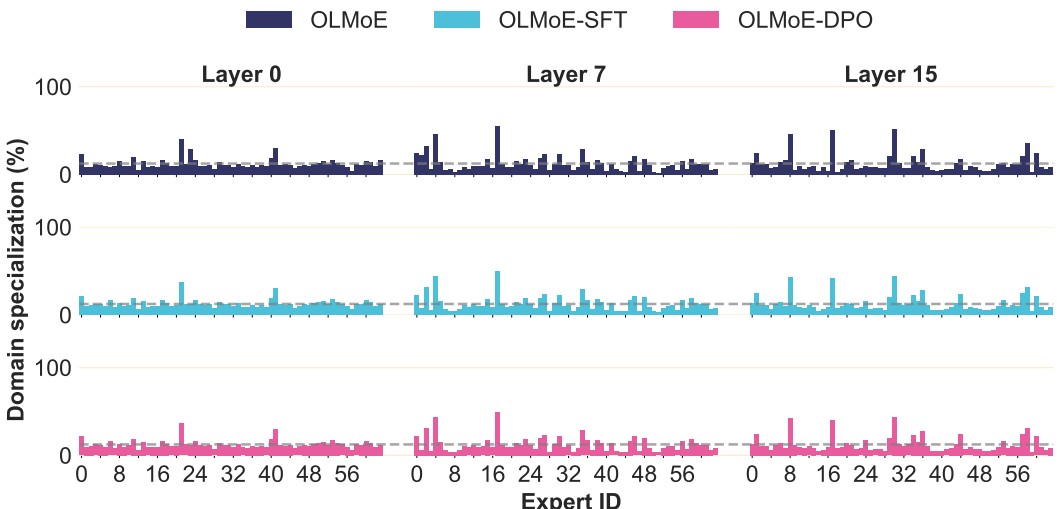

Figure H6: **Load imbalances in selective layers after adaptation.** We visualize how often tokens from our instruction tuning dataset (§2) get routed to the 8 active experts out of the 64 total experts ($k = 1$ in Equation 7). Horizontal gray lines correspond to uniform routing (8/64=12.5% per expert). Although we run SFT and DPO without loss balancing loss (§B.3), we observe that the load distribution does not change substantially.

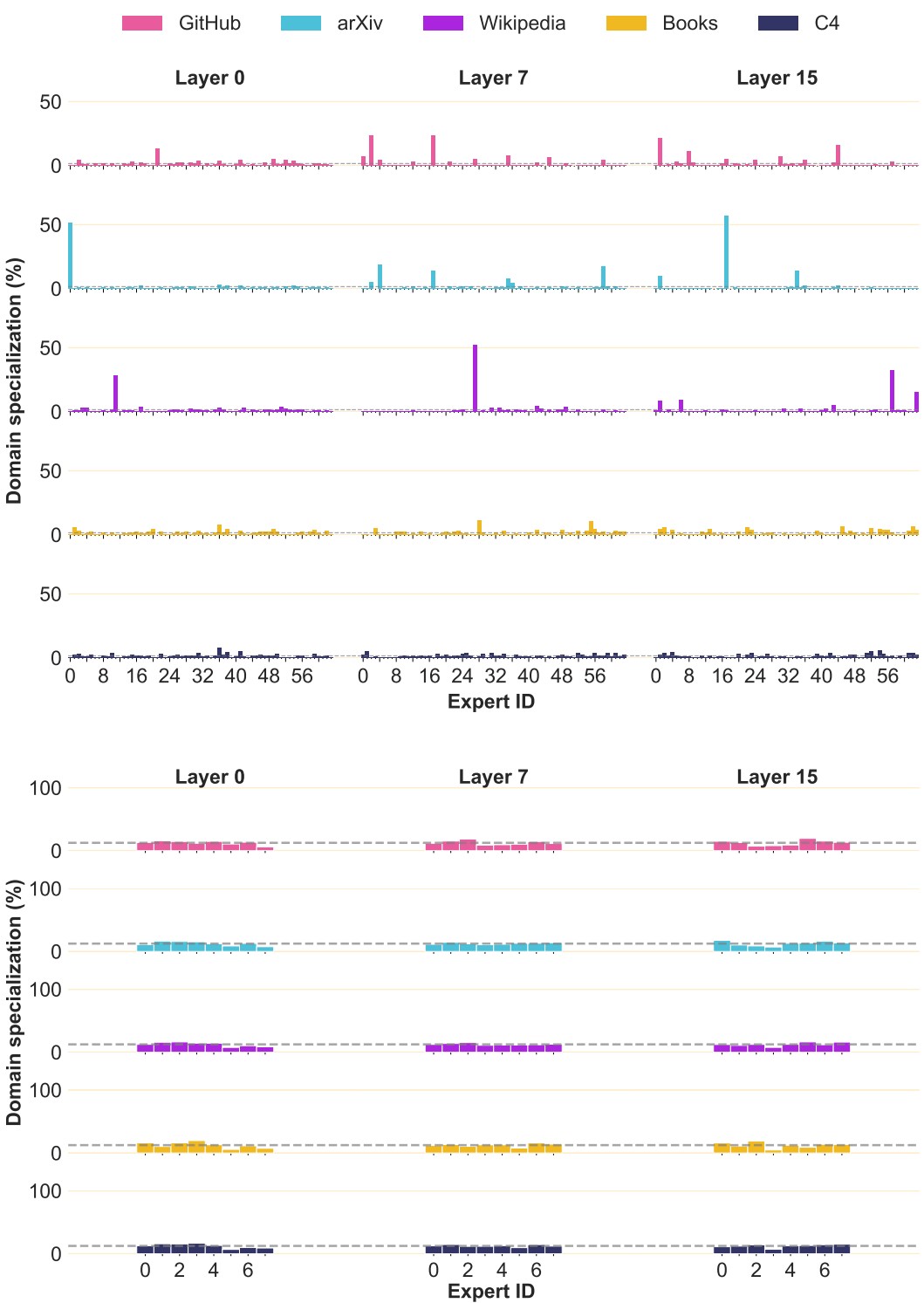

Figure H7: **Domain specialization of OLMOE-1B-7B (top) vs. Mixtral-8x7B (bottom) of the top-**1 **routed expert.** We visualize how often tokens from different domains get routed to the 64 (**OLMOE**) or 8 (Mixtral) experts at the end of pretraining. Unlike in Figure 7, here we only consider tokens routed to the top-1 expert ($k = 1$ in Equation 7). Horizontal gray lines correspond to uniform routing (1/64=1.56% per expert for **OLMOE-1B-7B** and 1/8=12.5% for Mixtral).

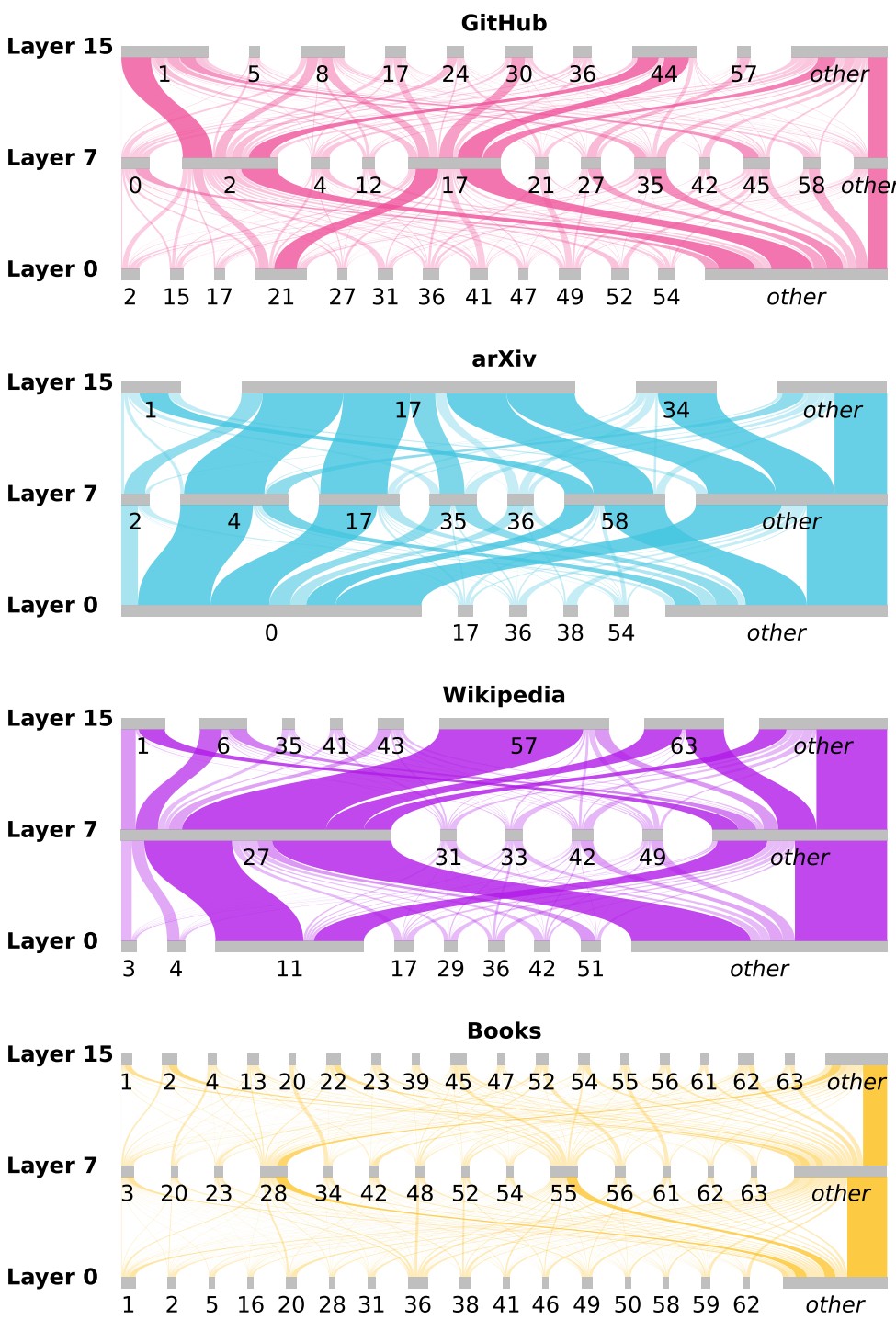

Figure H8: **OLMOE-1B-7B token routing across layers.** We visualize how often tokens from different domains get routed to a pair of experts across layers under top-1 routing, corresponding to Figure H7. The size of each rectangle is proportional to the total number of tokens an expert receives, while the flow between two experts shows the proportion of tokens routed to both experts. We only show experts that receive tokens 50% above random chance and use stronger coloring for larger flows. We observe some instances of cross-layer coordination between pairs of experts, e.g., expert 27 in layer 7 and expert 57 in layer 15 process a substantial fraction of Wikipedia tokens together. The flows between layers 0 → 7 and 7 → 15 are independent in this visualization.

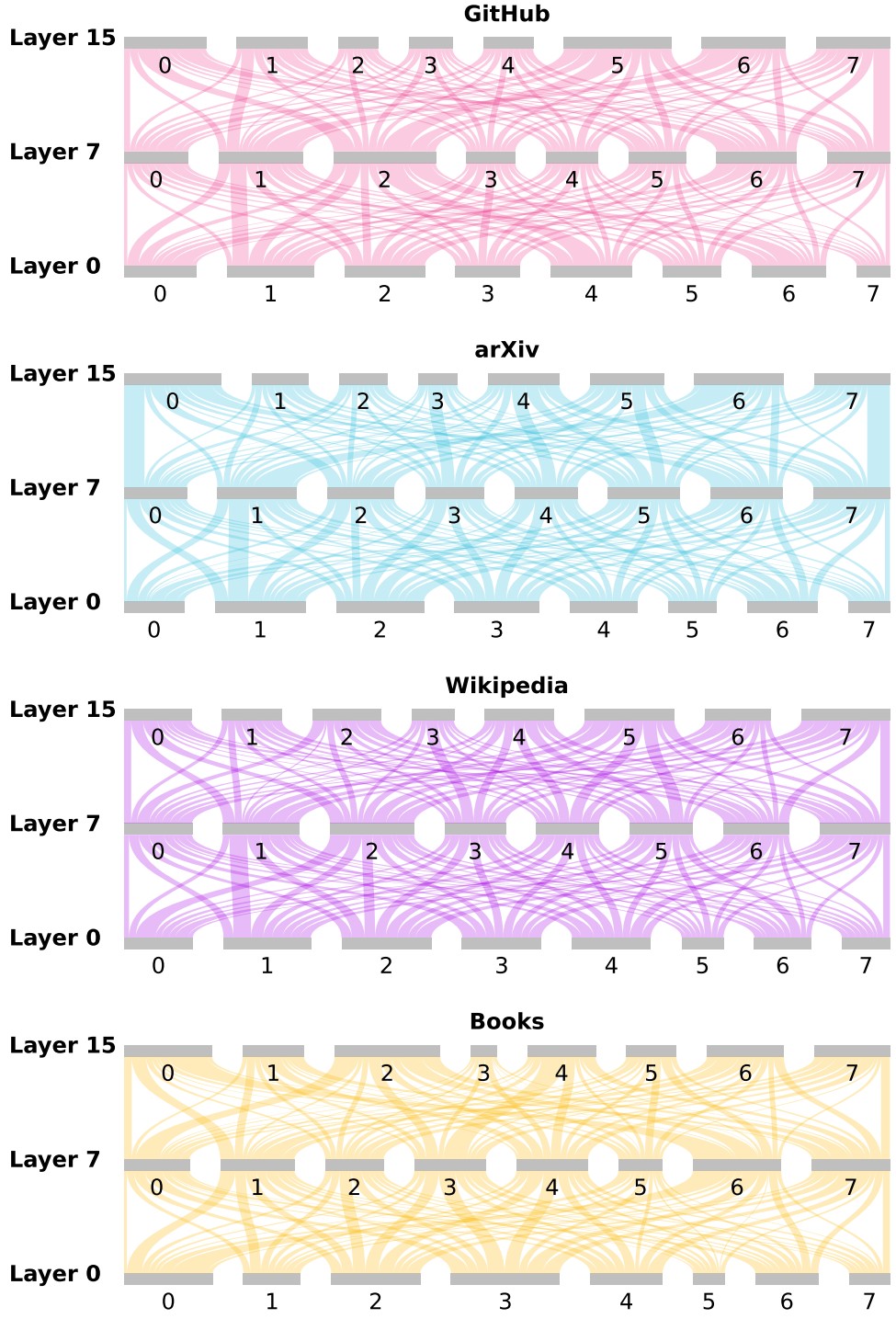

Figure H9: **Mixtral-8x7B token routing across layers.** We visualize how often tokens from different domains get routed to a pair of experts across layers under top-1 routing, corresponding to Figure H7. The size of each rectangle is proportional to the total number of tokens an expert receives, while the flow between two experts shows the proportion of tokens routed to both experts. The flows between layers 0 → 7 and 7 → 15 are independent in this visualization.

# I   ARTIFACTS

Table I1: **All artifacts released and used in this work.** We point from the name used for a given artifact in this work (e.g. Figure 1) to the URL where it can be obtained.

| Artifact | Public link |
|---|---|
| **OLMOE-1B-7B** | https://hf.co/allenai/OLMoE-1B-7B-0924 |
| **OLMOE-1B-7B-INSTRUCT** | https://hf.co/allenai/OLMoE-1B-7B-0924-Instruct |
| **OLMOE-1B-7B-SFT** | https://hf.co/allenai/OLMoE-1B-7B-0924-SFT |
| **OLMOE-MIX** | https://hf.co/datasets/allenai/OLMoE-mix-0924 |
| **SFT data** | https://hf.co/datasets/allenai/tulu-v3.1-mix-preview-4096-OLMoE |
| **KTO/DPO data** | https://hf.co/datasets/allenai/ultrafeedback_binarized_cleaned |
| **Code** | https://github.com/allenai/OLMoE |
| **Logs** | https://wandb.ai/ai2-llm/olmoe/reports/OLMoE-1B-7B-0924--Vmlldzo4OTcyMjU3 |
| BLOOM-7B | https://hf.co/bigscience/bloom-7b1 |
| DeepSeekMoE-3B-16B | https://hf.co/deepseek-ai/deepseek-moe-16b-base |
| DeepSeekMoE-3B-16B+chat | https://hf.co/deepseek-ai/deepseek-moe-16b-chat |
| DCLM-1B | https://hf.co/TRI-ML/DCLM-1B |
| DCLM-7B | https://hf.co/TRI-ML/DCLM-7B |
| Falcon-7B | https://hf.co/tiiuae/falcon-7b |
| Gemma2-3B | https://hf.co/google/gemma-2-2b |
| Gemma2-9B | https://hf.co/google/gemma-2-9b |
| JetMoE-2B-9B | https://hf.co/jetmoe/jetmoe-8b |
| JetMoE-2B-9B+SFT | https://hf.co/jetmoe/jetmoe-8b-sft |
| JetMoE-2B-9B+Chat | https://hf.co/jetmoe/jetmoe-8b-chat |
| Llama-7B | https://hf.co/huggyllama/llama-7b |
| Llama2-7B | https://hf.co/meta-llama/Llama-2-7b-hf |
| Llama3.1-8B | https://hf.co/meta-llama/Meta-Llama-3.1-8B |
| MPT-7B | https://hf.co/mosaicml/mpt-7b |
| Mistral-7B | https://hf.co/mistralai/Mistral-7B-v0.1 |
| Mixtral-8x7B | https://hf.co/mistralai/Mixtral-8x7B-v0.1 |
| OLMo-1B (0724) | https://hf.co/allenai/OLMo-1B-0724-hf |
| OLMo-7B (0724) | https://hf.co/allenai/OLMo-7B-0724-hf |
| OpenMoE-3B-9B | https://hf.co/OrionZheng/openmoe-8b |
| Pythia-7B | https://hf.co/EleutherAI/pythia-6.9b |
| Qwen1.5-3B-14B | https://hf.co/Qwen/Qwen1.5-MoE-A2.7B |
| Qwen1.5-3B-14B+Chat | https://hf.co/Qwen/Qwen1.5-MoE-A2.7B-Chat |
| StableLM2-2B | https://hf.co/stabilityai/stablelm-2-1_6b |
| TinyLlama-1B | https://hf.co/TinyLlama/TinyLlama_v1.1 |

# J   SELECTING THE NUMBER OF TOTAL AND ACTIVE PARAMETERS

In addition to what we mention in §4.1, there are three key reasons we select a configuration of 1B active parameters and 7B total parameters for **OLMOE-1B-7B**.

**Model training**   7B total parameters allow for full-parameter training on a single GPU. Specifically, our model can be trained on one 80GB VRAM GPU (e.g. A100 or H100) as it requires around 70GB of memory for training the model in 16-bit with an 8-bit optimizer (Anthony et al., 2023). This makes the model significantly more accessible to researchers who are often constrained by a single GPU and also bypasses the need for more complicated distributed training across multiple GPUs. A slightly larger model (e.g. JetMoE-2B-9B) may no longer fit under this setup.

**Model usage on laptops**   Laptops commonly have around 16GB of RAM, thus 7B parameters corresponding to 14GB in 16-bit precision perfectly fit into most laptop's RAM. With the speed of 1B parameters, the configuration of **OLMoE-1B-7B** could make it an ideal local assistant.

**Model usage on phones**   We have been able to run **OLMoE-1B-7B** on an iPhone by quantizing the model to 4-bit after which it requires around 3.5GB ($0.5 * 7$) of RAM. This is just below the 5GB RAM limit that is commonly imposed for an iOS app[18] leaving  1.5GB of RAM for other functionalities of the app.  Thanks to the 1B active params **OLMoE-1B-7B** runs very fast on smartphones; we were able to run it at 110 tokens/second on an iPhone 16. This enables applications that might not make sense with larger and slower models, such as having the model quickly read multiple long files and summarize them.

## K   LIMITATIONS AND FUTURE WORK

We highlight four key limitations with this release of **OLMoE-1B-7B**. We look forward to addressing these issues in future iterations of **OLMoE**.

**More parameters**   **OLMoE-1B-7B** has 7B total parameters out of which 1B are activated for each input token. This small size makes **OLMoE-1B-7B** very cheap to use, yet we demonstrate in this work that it outperforms much more expensive models (Figure 1). We provide further reasons for this precise configuration in Appendix J. However, using only 1B parameters for each input token also limits the capabilities of **OLMoE-1B-7B** as seen by its performance compared to models that use >7× more parameters, such as Llama3.1-8B in §3. While it may be possible that more parameters are not needed to match 8B models and beyond (Karpathy, 2024), in the short-term adding parameters is an easy way to improve the performance of **OLMoE**. Significantly adding parameters may, however, make dropless routing (Gale et al., 2022) as used in this work more challenging and may require expert parallelism (Lepikhin et al., 2020) with token dropping. We note that the DBRX model also uses dropless routing (Databricks, 2024; Gale et al., 2022) at a scale of 36B active and 132B total parameters. A different approach to more parameters could be allowing the model to utilize more than 1B parameters per input, possibly via recursion (Dehghani et al., 2019) or agentic workflows (Wang et al., 2024a; Yang et al., 2024b). Relatedly, changing the allocation of parameters to e.g. vocabulary versus non-vocabulary parameters is another avenue for improvement (Tao et al., 2024).

**More data**   We train **OLMoE-1B-7B** for 5 trillion tokens, however, some recent dense models train significantly longer, such as Llama 3 with 15 trillion tokens (Dubey et al., 2024). To the best of our knowledge, there has been no large MoE that has been overtrained (Gadre et al., 2024) as much as **OLMoE-1B-7B**. Specifically, taking the active parameters of **OLMoE-1B-7B**, our token multiplier (Gadre et al., 2024) is around 5,000 (5T / 1B). There are likely benefits to training even longer, but to what degree overtraining is effective for MoEs and how it differs from dense models still requires more research (Allen-Zhu & Li, 2024).

**Multimodal**   **OLMoE-1B-7B** is a text-only large language model, thus it cannot take inputs or produce outputs in other modalities like images or audio. This limits its utility for the large variety of multimodal use cases of such models (Huang et al., 2018; Su et al., 2020; Chen et al., 2020; Kiela et al., 2021; Muennighoff, 2020; Radford et al., 2022; Bai et al., 2023b; Driess et al., 2023; Dubey et al., 2024). There has been early work on open multimodal MoEs (Mustafa et al., 2022; Lin et al., 2024a; Li et al., 2024b; Shen et al., 2023b; McKinzie et al., 2024; Wu et al., 2024a) and we look forward to making future versions of **OLMoE** a part of that.

**Multilingual**   We pretrain **OLMoE-1B-7B** on a predominantly English corpus and exclusively evaluate on English tasks.  This may severely limit the usefulness of our model for research on non-English language models (Lovenia et al., 2024; Singh et al., 2024; Üstün et al., 2024; Enevoldsen et al., 2024; Son et al., 2024; Xiao et al., 2023). While there has been work on training language-specific LMs (Luukkonen et al., 2023; Faysse et al., 2024), it is more likely that as we add more data to build better future iterations of **OLMoE** we will mix in more non-English data due to data

---

[18]https://github.com/thebaselab/codeapp/issues/259

constraints (Muennighoff et al., 2023b). This may make future **OLMoE** models perform better in non-English languages.

## L OLMoE-1B-7B-0125

We introduced **OLMoE-1B-7B** in September 2024. In January 2025, we released a better model, **OLMoE-1B-7B-0125**, which we discuss here.

Table L1: DOLMINO composition and sampling distribution used for **OLMoE-1B-7B-0125**.

| Source | Total tokens | Source % | Mix % |
|---|---|---|---|
| Filtered DCLM | 752B | 6.85 | 50.2 |
| Decontaminated FLAN | 17.0B | 100 | 16.7 |
| StackExchange Q&A | 1.26B | 200 | 2.47 |
| peS2o | 58.6B | 16.7 | 9.52 |
| Wikipedia/Wikibooks | 3.70B | 100 | 3.57 |
| Dolmino Math | 10.7B | 200 | 17.5 |

For pretraining, **OLMoE-1B-7B-0125** uses the same data mix for the first stage of training. Following OLMo 2 (OLMo et al., 2024), we anneal this new model on a curated mix of high-quality sources. We sample this mix from the DOLMINO dataset,[19] a collection of high-quality web pages, academic content, question answering pairs, instruction data, and math problems. We use the same 100B tokens sample of DOLMINO used to anneal OLMo 2 13B; a summary of this dataset is in Table L1.

Table L2: **OLMoE-1B-7B-0924 and OLMoE-1B-7B-0125 on OLMES.** We bold the best performance. [†] indicates use of the MCF score, see Appendix D for evaluation details.

| OLMoE release | ARC_C | ARC_E | BoolQ | CSQA | HSwag | MMLU | OBQA | PIQA | SIQA | WinoG | Avg |
|---|---|---|---|---|---|---|---|---|---|---|---|
| **Sep 2024 (0924)** | 62.1[†] | 84.2 | 79.2 | **72.9** | 80.0 | 54.1[†] | 65.4[†] | **79.8** | 63.0[†] | 70.2 | 71.1 |
| **Jan 2025 (0125)** | **67.5**[†] | **84.4**[†] | **80.6** | 70.8 | **81.7** | **56.3**[†] | **69.6**[†] | 78.7 | **66.8**[†] | 70.6 | **72.7** |

We compare **OLMoE-1B-7B-0125** with **OLMoE-1B-7B** In Table L2. Overall, the new model is a notable improvement over the previous iteration being better on average (+1.6) and notable datasets like MMLU (+2.1).

Following this improved annealing setup, we adapt **OLMoE-1B-7B-0125** using the post-training from Tülu 3 (Lambert et al., 2025). This recipe represents an updated version of the one originally used for **OLMoE**. It features an improved SFT mix, better sampled DPO data, and a PPO step that leverages verifiers as for the model reward. We compare this new iteration using the evaluation setup from Tülu (which differs from other evaluations in this paper) in Table L3. After adaptation, the new model is significantly better, with a 10-point gain on the benchmark average.

The new models and datasets are freely available on the Hugging Face hub.[20] For more information about this release, we refer to its announcement on Ai2's website.[21]

---

[19]huggingface.co/datasets/allenai/dolmino-mix-1124

[20]hf.co/collections/allenai/olmoe-january-2025-67992134f9ebea0a941706ca

[21]allenai.org/blog/olmoe-app

Table L3: **OLMOE-1B-7B-0924 and OLMOE-1B-7B-0125 after adaptation.** We bold the best performance.

| **Skill** | **Benchmark**(eval) | **OLMOE-1B-7B-0924** | | **OLMOE-1B-7B-0125** | | |
|---|---|---|---|---|---|---|
| | | **+SFT** | **+DPO** | **+SFT** | **+DPO** | **+RLVR** |
| | Avg. | 39.7 | 39.8 | 46.6 | 49.3 | **49.8** |
| Knowledge | MMLU(0 shot, CoT) | 54.3 | 54.6 | **55.3** | 54.9 | 55.1 |
| | PopQA(15 shot) | **21.0** | 20.6 | 20.1 | 19.7 | 19.8 |
| | TruthfulQA(6 shot) | 44.7 | 49.1 | 45.5 | 50.0 | **50.6** |
| Reasoning | BigBenchHard(3 shot, CoT) | 36.6 | 36.8 | 37.3 | 37.4 | **38.6** |
| | DROP(3 shot) | 34.7 | 34.5 | **48.6** | 48.4 | 47.9 |
| Math | MATH(4 shot CoT, Flex) | 8.2 | 8.2 | **21.4** | 20.4 | **21.4** |
| | GSM8K(8 shot, CoT) | 42.5 | 47.4 | 55.7 | 64.6 | **72.4** |
| Coding | HumanEval(pass@10) | **63.7** | 63.0 | 62.6 | 61.9 | 62.3 |
| | HumanEval+(pass@10) | 57.4 | 58.9 | 55.7 | **57.6** | 54.4 |
| IF & chat | IFEval(prompt loose) | 41.2 | 45.3 | 56.6 | 65.6 | **66.4** |
| | AlpacaEval 2(LC % win) | 6.4 | 7.5 | 5.8 | **19.5** | 18.0 |
| Safety | Safety(6 task avg.) | 65.8 | 51.4 | **94.5** | 91.4 | 90.4 |

