# OpenReview forum: "OLMoE: Open Mixture-of-Experts Language Models"
_ICLR.cc/2025/Conference — ICLR 2025 Oral_

### Official Review · Reviewer_vUvg · 2024-10-29

**Soundness:** 4
**Presentation:** 4
**Contribution:** 4
**Rating:** 10
**Confidence:** 5

**Summary:**

This work is devoted to sharing the insights, data, and checkpoints of a series of MoE LLMs. The model achieved promising results on various benchmarks as a fully open model family.

**Strengths:**

1) There is no doubt that training MoE LLMs is challenging. This work offers a couple of important takeaways about how to train a good MoE LLMs, which is very helpful to the community.
2) The presentation is very clear. For instance, the Table 1 delivers many key designs clearly at the early section of the paper.
3) The model performance is good as well. As shown in Table 2 and 3, the model performs competitive with dense open models and partially open models (e.g. Qwen, Deepseek).
4) The Analysis in Section 5 is informative, which greatly help readers and authors to understand how is the model working. This can also greatly speedup the growth of the community.

**Weaknesses:**

1) Although the model has been relatively large, it is still much smaller than the SoTA MoE LLMs. I understand it is hard to get enough training resource for a fully open projects.

**Questions:**

1) What do you think about the necessity of expert parallelism? This model used dropless MoE, so it anyway will be unbalanced when using expert parallelism during training and inference. Without expert parallelism, it is still okay when the model is small. However, if we are aiming at a very large model, which has very large experts even if we are using the "fine-grained MoE", the expert parallelism would still be required? So how can we handle the token drop problem in this case?

---

> ### Author Response · Authors · 2024-11-15
>
> Thank you for your detailed review and we are glad you find the analysis section informative — We indeed hope that this lowers the barriers of entry to MoEs and helps the growth of our community.
>
> **Regarding the model size being smaller than some other MoEs:** We fully agree with your point and also mention it in our limitations section (Appx K). Larger models are more resource-intensive - we hope that by fully openly releasing OLMoE, we enable the community to build on it and scale up even further in the future (something we are also currently investigating!).
>
> **Regarding the necessity of expert parallelism:** Great question — As you mention, we do not need expert parallelism at these small scales but it may be required at larger scales. The DBRX model with 36B active and 132B total parameters by Databricks [1] was also trained with dropless routing similar to OLMoE, so we should be able to scale up to that size at least without having to drop tokens. Another avenue could be switching to expert choice routing which does not drop tokens regardless of the scale - While we found it to perform worse in the paper, we hypothesize that it may perform better in multimodal settings. We have added a note on expert parallelism in Appendix K and updated the paper, thanks again for raising this.
>
> [1] Databricks. Dbrx, 2024. URL https://www.databricks.com/blog/introducing-dbrx-new-state-art-open-llm.

---

### Official Review · Reviewer_PJU8 · 2024-11-03

**Soundness:** 4
**Presentation:** 4
**Contribution:** 3
**Rating:** 8
**Confidence:** 2

**Summary:**

This paper presents a mixture-of-experts (MoE) LLM model called OLMoE that has 1B active parameters and 7B total parameters. The OLMoE model Pareto dominates many state-of-the-art models in the performance vs. active parameters space. The paper explores and presents insights on what is optimal in the design-space of MoE parameters and present analysis of routing behavior in MoEs.

**Strengths:**

1) Strong empirical results with state-of-the-art performance for 1B active parameters.
2) Good exploration of the MoE design space which forms a good guide for MoE model design.
3) Novel analysis of routing behavior in MoE models during training and inference.
4) This is the only MoE model where the model weights, code, data and checkpoints are openly available and thus the work is entirely reproducible.

**Weaknesses:**

1) Other state-of-the art MoE models in related works are not exactly in the same parameter count configuration (1B/7B) so an exact comparison cannot be made to this model's performance.
2) Most of the design choices and training choices are based on prior work and the novelty is more in the design space exploration and analysis of routing behavior.

**Questions:**

The work is well presented and possible suggestions for improvements are addressed in the future work section.

---

> ### Author Response · Authors · 2024-11-15
>
> Thank you for your review and highlighting the strong results and reproducibility aspects.
>
> Regarding your Weaknesses:
>
> **No other models available with the same configuration:** Thanks for bringing this up; indeed, we could not find state-of-the-art MoE models matching our configuration.  One of the motivating factors for our work is the lack of open models with this combination of active/total parameters. Specifically, we chose this configuration with three reasons in mind:
>
> **1. Model training:** 7B total parameters allow for full-parameter training on a single GPU. Specifically, our model can be trained on one 80GB VRAM GPU (e.g. A100 or H100) as it requires around 70GB of memory for [training the model in 16-bit with an 8-bit optimizer](https://blog.eleuther.ai/transformer-math/). This makes the model significantly more accessible to researchers who are often constrained by a single GPU and also bypasses the need for more complicated distributed training across multiple GPUs. A slightly larger model (e.g. JetMoE-2B-9B [1]) may no longer fit under this setup.
>
> **2. Model usage on laptops:** Laptops commonly have around 16GB of RAM, thus 7B parameters corresponding to 14GB in 16-bit precision perfectly fit into most laptop's RAM. With the speed of 1B parameters, the configuration of OLMoE-1B-7B could make it an ideal local assistant.
>
> **3. Model usage on phones:** We have been able to run OLMoE-1B-7B on an iPhone by quantizing the model to 4-bit after which it requires around 3.5GB. This is just below the ~5GB RAM limit that is [commonly imposed for an iOS app](https://github.com/thebaselab/codeapp/issues/25) leaving ~1.5GB of RAM for other functionalities of the app. Thanks to the 1B active params OLMoE-1B-7B runs very fast on smartphones; we were able to run it at 110 tokens/second on an iPhone 16. This enables applications that might not make sense with larger and slower models, such as having the model quickly read multiple long files and summarize them.
>
> To compensate for the lack of a model with a similar configuration, we compare with larger models like JetMoE (2B/9B) [1] and Qwen (3B/14B) [2] in the paper, state-of-the-art models for their respective parameter configurations. We find that OLMoE is competitive with these larger models, thus we would expect OLMoE to significantly outperform a similar model to theirs with a 1B/7B configuration. We have added this discussion on the 1B/7B configuration to the paper in a new Appendix section (J). We thank the reviewer for providing an opportunity to reflect on this important motivating aspect of our work!
>
> **Design choices based on prior work:** Thanks for highlighting this – our work heavily builds on important prior work in the field. The two areas of novelties that you mention include many new findings such as for “design space exploration”: the stronger performance of dropless token choice [3] over expert choice [4] and the limits of sparse upcycling [5] among others; and for “routing”: new measures for analysis including expert co-activation, specialization metrics, and router saturation. In addition to these two areas, we think that there are two more with novelties:
>
> **1. Some design choices themselves and their combination:** Some of our design choices are new (e.g. our use of a truncated normal initialization for an MoE; the specific dataset we mix), as well as some combinations are new (e.g. using both finegrained experts and router z-loss in one model). Overall, this leads to a novel and reproducible recipe for training a state-of-the-art MoE model.
>
> **2. Openness:** All aspects of OLMoE being fully open-source including 244 checkpoints is also a first among MoEs to the best of our knowledge. We hope this will enable more research into MoEs by lowering the barriers of entry for researchers!
>
>
>
> [1] Yikang Shen, Zhen Guo, Tianle Cai, and Zengyi Qin. Jetmoe: Reaching llama2 performance with 0.1m dollars, 2024. URL https://arxiv.org/abs/2404.07413.
>
> [2] Qwen Team. Qwen1.5-moe: Matching 7b model performance with 1/3 activated parameters”, February 2024b. URL https://qwenlm.github.io/blog/qwen-moe/.
>
> [3] Trevor Gale, Deepak Narayanan, Cliff Young, and Matei Zaharia. Megablocks: Efficient sparse training with mixture-of-experts, 2022. URL https://arxiv.org/abs/2211.15841.
>
> [4] Yanqi Zhou, Tao Lei, Hanxiao Liu, Nan Du, Yanping Huang, Vincent Zhao, Andrew Dai, Zhifeng Chen, Quoc Le, and James Laudon. Mixture-of-experts with expert choice routing, 2022. URL https://arxiv.org/abs/2202.09368.
>
> [5] Aran Komatsuzaki, Joan Puigcerver, James Lee-Thorp, Carlos Riquelme Ruiz, Basil Mustafa, Joshua Ainslie, Yi Tay, Mostafa Dehghani, and Neil Houlsby. Sparse upcycling: Training mixture-of-experts from dense checkpoints, 2023. URL https://arxiv.org/abs/2212.05055.

---

> > ### Author Response · Authors · 2024-11-23
> > **Follow-up**
> >
> > Dear Reviewer,
> >
> > We'd appreciate it if you'd let us know if our response has addressed your concerns.
> >
> > Thanks!

---

> > > ### Comment · Reviewer_PJU8 · 2024-11-23
> > >
> > > Thank you for the detailed insights. I do not have any further concerns. I did not consider the weaknesses with this work to be major to begin with so am keeping the "accept, good paper" score. However, I do appreciate your response and believe that it helps me and other reviewers understand the novelty of this work better.

---

### Official Review · Reviewer_7Ufw · 2024-11-09

**Soundness:** 4
**Presentation:** 4
**Contribution:** 4
**Rating:** 8
**Confidence:** 3

**Summary:**

This paper introduces OLMoE, a fully open, state-of-the-art language model built on a sparse Mixture-of-Experts (MoE) architecture. The authors conducted extensive experiments to validate the effectiveness of the proposed method, including evaluations after pre-training and adaptation phases. Additionally, they explored key design choices within the MoE framework, examining factors like expert granularity, routing strategies. Their analyses provided valuable insights into MoE, including router saturation, expert co-activation, and domain/vocabulary specialization.

**Strengths:**

- The writing in this paper is clear and easy to follow.
- The paper advances MoE research by providing a fully open-sourced, state-of-the-art MoE architecture, which is beneficial for the research community.
- The paper presents a thorough analysis of key design choices in MoE, offering valuable guidance on building high-performance MoE models.
- The analysis is insightful, with discussions on phenomena such as router saturation and expert co-activation providing fresh perspectives and meaningful implications for the field.

**Weaknesses:**

I have a question regarding the experimental results: were the model parameters quoted directly from the original paper for the results shown in Table 2? For instance, in the original paper, OpenMOE’s activation parameter count is reported as 2.1B, whereas Table 2 shows an activation parameter count of 2.9B for OpenMOE. I recommend that the authors carefully verify the accuracy of these values.

**Questions:**

See Above.

---

> ### Author Response · Authors · 2024-11-15
>
> Thanks a lot for your detailed review - we are glad you found our analysis insightful and delivering fresh perspectives!
>
> **Regarding your question about whether model parameters are quoted from the original papers:** Yes, we intended to quote parameters directly from their papers, but there is indeed a typo in Table 2 and it should be 2.6B instead of 2.9B for OpenMoE. This number comes from Appendix B, Table 11 in the OpenMoE paper [1] and corresponds to their active parameters with embedding. 2.1B is their active parameters without embedding, but since we include the embedding parameters in the active parameters of all other models, we also do so for OpenMoE. In Figure 1, we placed OpenMoE correctly corresponding to 2.6B. We have fixed the typo in Table 2 to 2.6B and revised the submission. We have also double-checked all other numbers. Thanks a lot!
>
> [1] Fuzhao Xue, Zian Zheng, Yao Fu, Jinjie Ni, Zangwei Zheng, Wangchunshu Zhou, and Yang You. Openmoe: An early effort on open mixture-of-experts language models, 2024. URL https://arxiv.org/abs/2402.01739.

---

> > ### Author Response · Authors · 2024-11-23
> > **Follow-up**
> >
> > Dear Reviewer,
> >
> > We'd appreciate it if you'd let us know if our response has addressed your concerns.
> >
> > Thanks!

---

> > > ### Comment · Reviewer_7Ufw · 2024-11-25
> > > **Response to Authors**
> > >
> > > Thank you for your detailed response. All my concerns have been well addressed.

---

### Author Response · Authors · 2024-11-15

We thank all reviewers for their detailed reviews. We have made the following changes to the paper and uploaded a new revision:

1. We have fixed a typo in the active parameter count of OpenMoE in Table 2 thanks to feedback from reviewer 7Ufw.
2. We have added a discussion of the advantages of the specific OLMoE configuration (1B/7B) in a new Appendix section (J), including quantization experiments we ran achieving 110 tokens/second generation speed on an iPhone with OLMoE, in response to reviewer PJU8.
3. We have added notes on expert parallelism and how the community can scale up OLMoE in Appendix K to address the question raised by reviewer vUvg.

Overall, we are glad reviewers have found the paper to provide fresh and novel insights. Reviewers have also pointed to the open nature and reproducibility of the work, as well as its artifacts. Building on these, we hope that together with the broader community, we can further advance the state of fully open language models.

---

### Meta-Review · Area_Chair_h9JQ · 2024-12-19

**Metareview:**

The paper introduces a state-of-the-art language model leveraging a sparse Mixture-of-Experts (MoE) architecture. It presents novel findings on MoE training, defines and analyzes new routing properties showing high specialization in their model, and open-sources all aspects of their work, including model weights, training data, code, and logs.

The paper provides significant contributions to the field of MoE models, offers valuable insights, and is fully open-source, making it a valuable resource for the research community. Based on the reviews and the authors' thorough responses, I recommend acceptance.

**Additional Comments On Reviewer Discussion:**

The authors addressed all concerns raised by the reviewers, including correcting the typo in the model parameters and adding discussions on the advantages of their specific configuration and expert parallelism.

---

### Decision · Program_Chairs · 2025-01-22

Accept (Oral)